# TopoNets: High performing vision and language models with brain-like topography

**Mayukh Deb**[1,2], **Mainak Deb**[3], **N. Apurva Ratan Murty**[1,2]
[1]Cognition and Brain Science, School of Psychology, Georgia Tech
[2]Center for Excellence in Computational Cognition, Georgia Tech
[3]Independent Contributor
{mayukh, ratan}@gatech.edu

## Abstract

Neurons in the brain are organized such that nearby cells tend to share similar functions. AI models lack this organization, and past efforts to introduce topography have often led to trade-offs between topography and task performance. In this work, we present *TopoLoss*, a new loss function that promotes spatially organized topographic representations in AI models without significantly sacrificing task performance. TopoLoss is highly adaptable and can be seamlessly integrated into the training of leading model architectures. We validate our method on both vision (ResNet-18, ResNet-50, ViT) and language models (GPT-Neo-125M, NanoGPT), collectively *TopoNets*. TopoNets are the highest performing supervised topographic models to date, exhibiting brain-like properties such as localized feature processing, lower dimensionality, and increased efficiency. TopoNets also predict responses in the brain and replicate the key topographic signatures observed in the brain's visual and language cortices, further bridging the gap between biological and artificial systems. This work establishes a robust and generalizable framework for integrating topography into AI, advancing the development of high performing models that more closely emulate the computational strategies of the human brain. Our project page: https://toponets.github.io

## 1 Introduction and Related Work

Neurons in the brain are not tossed around haphazardly; they're spatially organized such that nearby cells perform similar functions (Barlow, 1986; Rakic, 1988; Eickhoff et al., 2018; Krubitzer, 2009). *Topographic organization* is a core feature of brains (Geschwind & Rakic, 2013; Arcaro & Livingstone, 2024). In visual cortex, this organization is evident in micro-scale pinwheel patterns for orientation selectivity(Maldonado et al., 1997; Bonhoeffer & Grinvald, 1991), in macro-scale category-selective regions for faces (Kanwisher & Yovel, 2006; Kanwisher et al., 2002), bodies (Downing et al., 2001), scenes (Epstein et al., 1999) etc and in large-scale organizational biases for real-world shape, size and animacy (Konkle & Caramazza, 2013; Konkle & Oliva, 2011). Beyond vision, in the language cortex, recent studies have also identified neurons with distinct temporal integration windows (Hasson et al., 2008; Lerner et al., 2011; Regev et al., 2024). Unlike the brain, most artificial neural network (ANN) models lack any systematic organization of units. In this work, we introduce a new brain-inspired inductive bias, *TopoLoss*, that can be integrated into the training of most current ANN architectures, including convolutional networks and transformers. The resulting models, *TopoNets*, bring brain-like topography to AI, yielding high-performing models with localized, low-dimensional, and efficient representations — much like the brain (Figure 1).

Inducing topography into artificial neural networks (ANNs) has proven to be challenging and two main strategies have emerged. The first, *post-hoc topography*, involves re-organizing units in pretrained models using methods like self-organizing maps (Doshi & Konkle, 2023; Zhang et al., 2021; Kohonen, 1997). The resulting models exhibit topographic signatures, but the underlying representations remain unchanged from the original model. Consequently, the functional advantages of topography, such as reduced dimensionality and increased efficiency, are not realized. The second strategy, *jointly-optimized topography*, incorporates an additional topographic loss during model

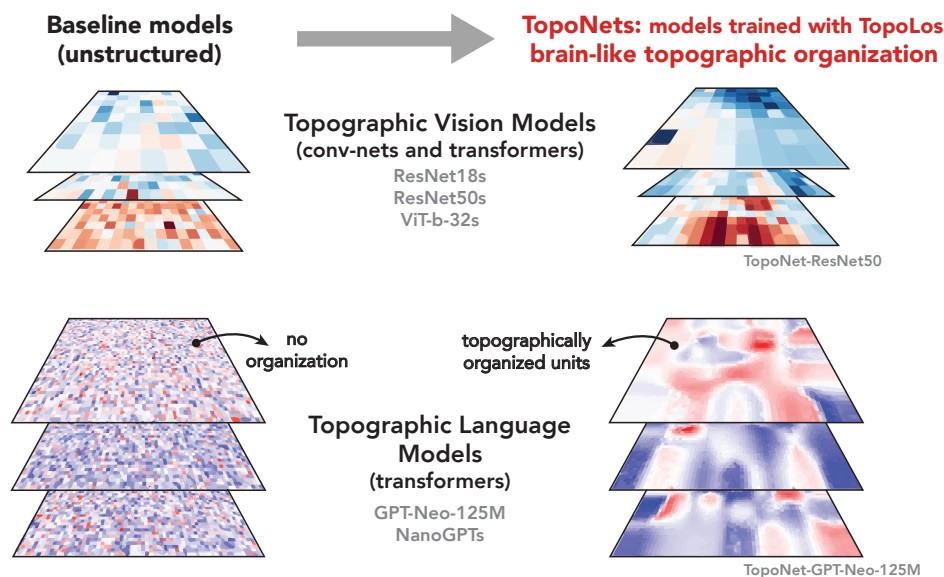

Figure 1: **Towards high performing topographic vision and language models (TopoNets).** Schematic shows transformation from unstructured baseline models (left) to organized topographic representations (right) for vision (top) and language models (right). The stacked maps are 3 representative layers (early, mid, and late) of the model.

training. These models induce topography by (1) explicitly matching the brain's spatial correlation structure (Lee et al., 2020; Margalit et al., 2024), (2) imposing distance-dependent constraints (Blauch et al., 2022; Qian et al., 2024), or (3) encouraging information redundancy (Keller et al., 2021). These approaches suffer a significant tradeoff: the ability of the model to learn task-relevant representations is often compromised. They perform poorly on engineering metrics (like performance on ImageNet) and/or show diminished capacity to predict brain data. Also most prior work in this space has focused exclusively on vision models and the one attempt ((Binhuraib et al., 2024)) at imparting topography to language models focused on the self-attention maps of BERT, which resulted in only modest topographic organization in the output space. To summarize, no unified strategy currently exists for applying topography across ANN algorithms (i.e., convolutional nets and transformers) and domains (vision and language) that can deliver *high-performing* models together with the *functional benefits of topography*.

Here, we set out to design a general inductive model bias to recapitulate signatures of brain-like topography and its computational benefits without sacrificing model accuracy. To achieve this, it was important to understand *why* and *how* topography arises in the first place. Theoretical work in neuroscience suggests that one of the primary evolutionary pressures on the brain is metabolic efficiency — not only in terms of minimizing wiring length of neurons (Kaas, 1997), but also in managing the vast network of potential neural connections (Katz & Shatz, 1996; Chklovskii et al., 2002; Chklovskii & Koulakov, 2004). The brain addresses this challenge through synaptic pruning. Early in development, the brain forms an excess of synaptic connections, which are then systematically reduced over time based on activity-dependent mechanisms (Kaas, 1997; Faust et al., 2021; Riccomagno & Kolodkin, 2015; Schulz & Reggia, 2005). This pruning process retains only the most necessary connections, optimizing for the efficiency of the neural network. Our topographic loss incorporates ideas about synaptic pruning into its design.

Our study makes the following contributions: A) We introduce *TopoLoss*, a new inductive bias that generalizes across model architectures (convolutional networks and transformers) and domains (vision and language). (B) We show that TopoNets, our suite of supervised topographic models, outperform previous topographic models on ImageNet performance and predictions on brain data (as on BrainScore) while maintaining similar levels of topography. (C) TopoNets provide clear evidence resolving theoretical claims about the role of topography in creating low-dimensional feature

representations. (D) We show that TopoNets demonstrate improved model efficiency. (E) TopoNets replicate topographic signatures observed in vision and language cortex in the brain.

## 2 METHODS

### 2.1 DEFINING THE CORTICAL SHEET

Our first task was to define a 2D sheet (See Appendix A.1 for a discussion of topography in 2D versus 3D spaces) where we could apply our topographic loss (*TopoLoss*). To demonstrate that TopoLoss generalizes across different domains, we applied it to both language and vision models. For **language**, we trained GPT-Neo-125M models (Black et al., 2021) on the Wikipedia Dataset (Wikimedia Foundation) and NanoGPT models Karpathy (2022) on 10 billion tokens from FineWeb-Edu (Lozhkov et al., 2024). For **vision**, we trained topographic ResNet-18 to allow comparisons with previous topographic models, and ResNet-50 (He et al., 2016) and ViT-b32 Dosovitskiy (2020) models to further evaluate the generalization of TopoLoss to larger models and architectures. All vision models were trained on a supervised 1000-way classification task on ImageNet (Deng et al., 2009). Together these language and vision models allowed us to robustly evaluate TopoLoss across varied model architectures and domains.

**Cortical sheet in Transformers (language models and ViTs)**: For a linear layer with $i$ input units and $o$ output units, we reshape its weight matrix $\mathbf{W} \in \mathbb{R}^{o \times i}$ to a cortical sheet $\mathbf{C} \in \mathbb{R}^{h \times w \times d}$. In this setup, the area of the sheet $(h \times w)$ corresponds to the number of output units $(o)$ and the depth $(d)$ corresponds to the number of input units $(i)$. To maximize the number of neighbors for each element in the cortical sheet, we chose $h$ and $w$ to be as close to each other as possible, thereby minimizing the perimeter. Each"element" in this cortical sheet now represents the weights associated with a single output unit (or "neuron") in the original linear layer.

**Cortical sheet in Convolutional Models**: For a convolutional layer with $c_{\text{input}}$ input channels and $c_{\text{output}}$ output channels, and a kernel-size of $k \times k$, we project its weight tensor $\mathbf{W} \in \mathbb{R}^{c_{\text{output}} \times c_{\text{input}} \times k \times k}$ onto a cortical sheet $\mathbf{C} \in \mathbb{R}^{h \times w \times d}$, where the area corresponds to the number of output channels $(h \times w)$ and the depth is defined as $d = c_{\text{input}} \times k \times k$. As in previous work (Qian et al., 2024), we arranged the model units (convolutional kernels) on a 2D cortical sheet. A more detailed explanation is provided in the appendix A.2.

### 2.2 INDUCING TOPOGRAPHY (TOPOLOSS)

The second step introduces the TopoLoss to the reshaped cortical sheet, promoting topographic organization. We achieve this by maximizing the cosine similarity between the original cortical sheet and its blurred version. This suppresses the high-frequency noise, leaving behind only the most important and meaningful information. This idea was motivated by synaptic pruning in the brain, which eliminates noisy (high frequency) neural connections, refining the biological network's structure (although note that we do not explicitly *remove* any weights here). The blurring of a 2D signal $X \in \mathbb{R}^{h \times w}$ can be defined using a downsampling function $f_{down}$ and upsampling function $f_{up}$ a as follows:

$$\text{Blur}(X, \phi_h, \phi_w) = f_{\text{up}}\left(f_{\text{down}}\left(X, \frac{h}{\phi_h}, \frac{w}{\phi_w}\right), h, w\right) \tag{1}$$

Here $\phi_h$ and $\phi_w$ are the downsampling factors along height and width dimensions (both set to 3). To encourage smoothness in the cortical sheet $C^{h \times w \times d}$ we maximize the cosine similarity between $C$ and its blurred version $C'$ across cortical sheet layers maps. This process smoothens the representations and encourages topographic organization. The TopoLoss is defined as:

$$\mathcal{L}_{\text{topo}} = -\frac{1}{N} \sum_{i=1}^{N} \frac{C_i \cdot C_i'}{\|C_i\| \|C_i'\|}$$

This TopoLoss is integrated with the original training loss $\mathcal{L}_{\text{training}}$ as follows:

$$\mathcal{L}_{\text{total}} = \mathcal{L}_{\text{training}} + \tau(\mathcal{L}_{\text{topo}})$$

Here $\tau$ is a scaling factor that controls the strength of the topographic effect: higher values encourage stronger topographic organization in the model.

**Vision Models**: We applied TopoLoss to every convolutional layer in the residual blocks (as (Qian et al., 2024)). All vision models were trained on a supervised 1000-way classification task using ImageNet. **1. ResNet-18** We trained 8 distinct ResNet-18 (He et al., 2016) models from scratch on the ImageNet (Deng et al., 2009) dataset across various topographic configurations: one baseline model (no topography), six TopoNets with different topographic scaling factors: $\tau = 0.5, 1, 5, 10, 20, 50$. Models were trained using the `ffcv` (Leclerc et al., 2023) training recipe. `ffcv` (Fast Forward Computer Vision) significantly accelerates model training by replacing traditional data loaders with an efficient binary format and leveraging multiprocessing and GPU-accelerated data augmentation to optimize data pipelines.

**ResNet-50:** We selected ResNet-18 to compare the performance of TopoNets with previous topographic approaches like TDANNs and LLCNN. However it has been demonstrated that ResNet-50 offers a richer visual representational basis for predicting brain responses (see: Ratan Murty et al. (2021); Lahner et al. (2024); McNeal et al. (2024); Khosla et al. (2022)). Hence we trained 3 additional ResNet-50 (He et al., 2016) models from scratch on ImageNet (Deng et al., 2009): one baseline model (no topography), two TopoNets with different topographic scaling factors ($\tau = 1, 30$).

**ViT-b-32** To demonstrate further generalizability beyond convolutional architectures for vision, we trained a Vision Transformer (Dosovitskiy, 2020) on the ImageNet dataset. We followed the recipe provided by TorchVision maintainers & contributors (2016) and applied TopoLoss with $\tau = 10$ on the last MLP module i.e the `mlp.3` module in each transformer block.

**GPT-Neo-125M:** We trained 5 GPT-Neo-125M (Black et al., 2021) models on the Wikipedia dataset (Wikimedia Foundation) with different scales of the topographic loss (baseline and $\tau$=1, 5, 10 and 50 respectively) . We applied TopoLoss to the c_fc layer of GPT-Neo. This choice was based on prior work by (Geva et al., 2020; 2022) that has suggested that the feed-forward modules in GPTs act as key-value memory modules storing world knowledge. The c_fc modules encode the persistent representations (in contrast to transient representations in the attention matrix) making it the theoretically grounded target for inducing topography.

**NanoGPT-125M** We trained 4 NanoGPT (Karpathy, 2022) models on 10 Billion tokens sampled randomly from the FineWeb-Edu dataset (Lozhkov et al., 2024) with different scales of the topographic loss (baseline and $\tau = 0.5, 1, 50$ respectively). TopoLoss was applied to the c_fc modules in each block (as explained above).

## 2.3 OTHER METRICS

**Effective Dimensionality:** Effective dimensionality was measured as described previously in (Margalit et al., 2024; Del Giudice, 2021; Elmoznino & Bonner, 2024).

$$\text{Effective Dimensionality} = \frac{\left(\sum_{i=1}^{n} \lambda_i\right)^2}{\sum_{i=1}^{n} \lambda_i^2}$$

$\lambda_i$ indicates the eigenvalues and $n$ the number of eigenvalues. This metric measures the *spread* of the eigenspectrum. For ResNets, we followed the procedure outlined in (Margalit et al., 2024). We chose 20,000 images from the ImageNet validation set calculated the effective dimensionality of the features for all the convolutional layers. For language models, we chose 8192 samples from the openwebtext dataset and measured dimensionality of the representations from the topographic (c_fc) layers.

**Smoothness** We measured topography, using smoothness score (as before, (Margalit et al., 2024)). Smoothness was defined as the difference between the highest and lowest correlation values from pairwise correlation versus distance plots.

**L1 unstructured pruning:** We impose sparsity by pruning a percentage of the smallest-magnitude weights. Specifically, we sort the weights in ascending order of their absolute magnitude and set the smallest $n\%$ of them to zero. To reduce the number of weights by a factor of $n$, we prune $(100 - \frac{100}{n})\%$ of the smallest weights. **Downsampling:** We downsample the topographic layers by first projecting them into the cortical space and then performing a downsample operation along the height and width dimensions of the cortical sheet. A detailed explanation of the downsampling operation and inference on such models can be found appendix A.5.

L1 unstructured pruning and downsampling were applied to progressively increase the degree of sparsity (ensuring that each sparsity level corresponds to the same effective parameter count) and evaluate the effect on model performance. For each sparsity level, we evaluated the model's performance (classification accuracy or perplexity) and reported the resulting performance difference from the baseline model (Figure 4).

**Estimating selectivity:** We collected layer-wise features in response to stimuli. Selectivity is then calculated using a standard method for estimating selectivity from these representations (e.g., (Margalit et al., 2024)):

$$t = \frac{\mu_c - \mu_o}{\sqrt{\frac{\sigma_c^2}{N_c} + \frac{\sigma_o^2}{N_o}}} \tag{2}$$

Where $\mu$, $\sigma$, $N$ denote the mean, standard deviation and the number of layerwise representations for the target category $c$ and other categories $o$. We used stimuli from previously published studies to identify the category-selective regions Stigliani et al. (2015) and regions with biases for real-world size and animacy (Konkle & Caramazza, 2013).

**Temporal window analyses:** We estimated the temporal integration window of every unit within GPT-Neo-125M using a recently developed method (Skrill & Norman-Haignere, 2024). Briefly, this approach employs a word-swapping paradigm, measuring the difference in response magnitude for the swapped sequences. The integration window is defined as the distance-dependent change in response magnitude across multiple sequences and word swaps. For detailed methodology, we refer readers to this important study. The key equation relevant to our work is as follows.

$$\theta_{\text{norm}}[\Delta] \approx c(\Delta + 1)^{-a} + (1 - c)e^{-b\Delta} \tag{3}$$

$\theta_{norm}[\Delta]$ is the normalized temporal integration window and $c$ is the convex combination parameter. Intuitively $c$ is the balancing knob that controls the relative influence of two different integration window shapes. $a$ represents the power-law component of the integration window. Higher values would indicate a relatively slower decline. $b$ is the exponential component of the integration window. Higher numbers would indicate a much more rapid decline. We followed exactly the same procedures outlined in the previous study to estimate these values.

## 3 RESULTS

### 3.1 TOPONETS ACHIEVE HIGH MODEL PERFORMANCE WITH COMPARABLE SPATIAL TOPOGRAPHY

How do TopoNets stack up against baseline models and other topography-inducing methods? We first tested vision models, specifically ResNet18 trained in a supervised manner on ImageNet. This architecture allows for direct comparison with previous work (Margalit et al., 2024; Qian et al., 2024). (Note: data for ITN (Blauch et al., 2022) and All-TNNs (Lu et al., 2023) are unavailable, as these architectures haven't been scaled to ImageNet). Model performance is presented against the amount of topography (smoothness, see (Margalit et al., 2024)) in Figure 2A. We find that (1) TopoNet-ResNet18 models achieved significantly higher accuracy on Imagenet (red dots) than LL-CNN (Imagenet trained supervised) and TDANN. TDANNs were trained using a self-supervised SimCLR objective, while TopoNets were trained in a supervised manner. To ensure fairer comparison with TDANNs (see A.3 for extended discussion), we trained an additional TopoNet with topography induced in similar locations as in TDANN (in 8 locations) and compared the *change* in performance from baseline (non-topographic). TDANNs exhibited a 6% drop in performance, whereas TopoNets showed only 3% drop. (2) TopoNets achieved comparable levels of topography (dashed gray vertical lines) as previous approaches. (3) TopoNets were trained for fewer training epochs (12% fewer than LLCNN-G ). Even the worst performing TopoNet-ResNet18 ($\tau = 50$) was significantly better than the previous best topographic model (25% drop for TopoNet-ResNet18-$\tau 50$ compared to 41% drop for LLCNN-G, from baseline). The model performance Pareto curve for TopoNet-ResNet18 model is shown as a black dashed line across levels of topography. Together these results indicate that models trained in a supervised manner with our new model inductive bias: TopoLoss (TopoNets) achieve substantially higher task performance than prior topographic models. TopoNets set a new standard of performance for supervised topographic ResNet18s.

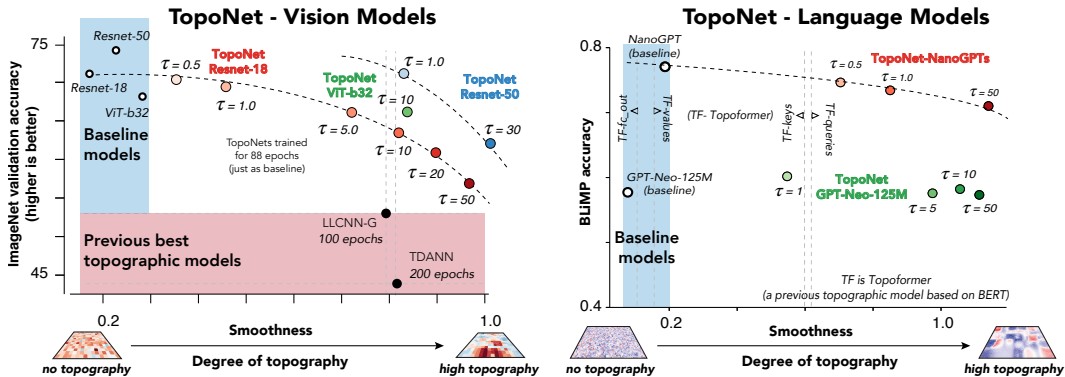

Figure 2: **TopoNets achieve higher model performance with comparable topography. A.** Estimated model topography (smoothness, x-axis) versus model performance (y-axis) for vision models (ResNet-18, ResNet-50, ViTs). The black filled dots with the dashed gray crosshairs indicate prior models. The dashed black lines indicate the pareto-curves for ResNet-18 and ResNet-50 models. **B.** Same as A, but for Language models (GPT-Neo-125M and NanoGPT). The y-axis here denotes the language model evaluation score on BLiMP. The dashed gray line indicates the reported topography from a prior study.

Next, we investigated whether we could develop even higher-performing TopoNets and extend the approach to transformer architectures. We trained 2 ResNet-50 models and one ViT-b32 model with TopoLoss. The model performance and topography measurements are shown for TopoNet-ResNet-50s and TopoNet-ViT-b32 as blue and green dots respectively in Figure 2. TopoNet-ResNet-50s and TopoNet-ViT both outperformed TopoNet-ResNet-18s at similar levels of topography. Notably, our TopoNet-ResNet50-$\tau 1$ achieved comparable performance as the baseline ResNet-18 model, while exhibiting comparable levels of measured topography as prior topographic models.

Does TopoLoss generalize to language models? To investigate this question, we trained GPT-Neo-125M (with and without TopoLoss) on Wikipedia. To demonstrate further scalability, we also trained NanoGPT models on 10 billion tokens from the FineWeb-Edu dataset. All models were evaluated on a common evaluation measure: BLiMP (Warstadt et al., 2020). Our findings revealed that (1) TopoNets were comparable to the baseline (non-topographic model) for GPT-Neo-125M and were close to baseline for the scaled up NanoGPT models (Figure 2B). (2) Most TopoNets achieved higher levels of topography than Topoformers (BERT trained on IMDB (Binhuraib et al., 2024)) even in the layers where topography was explicitly implemented (attention Q,K,V, Figure 2B). Together these results demonstrate that TopoLoss can generalize across different model architectures (convolutional nets and transformers) and domains (vision and language). The resulting models, TopoNets, significantly outperform previous isolated efforts in vision or language alone.

## 3.2 TOPOGRAPHY, NOT MODEL PERFORMANCE, DRIVES DIMENSIONALITY REDUCTIONS IN TOPONETS. EVIDENCE ACROSS MODEL ARCHITECTURES AND DOMAINS

Prior theoretical work has suggested that the brain's topography may affect non-topographic aspects of learned representations, such as the effective dimensionality (Durbin & Mitchison, 1990; Swindale, 1996). Effective dimensionality is lower when neurons are similar to each other and higher when they are independent. Studies show that 1) effective dimensionality increases with model depth and training, and 2) models with lower dimensionality better predict responses in high-level visual cortex (Elmoznino & Bonner, 2024). However, recent work (Qian et al., 2024) has can doubts on this observation. Specifically it is unclear whether the reduction in dimensionality is driven by lower model performance (Hypothesis 1) or because of topography itself (Hypothesis 2). TopoNets finally allow us to test these competing hypotheses more precisely across model architectures and domains.

We measured model dimensionality using a standard approach from previous studies (Del Giudice, 2021; Elmoznino & Bonner, 2024) and examined its relationship with model accuracy (Hypothesis 1) and topography (Hypothesis 2) across both vision (ResNet-18, ResNet-50) and language mod-

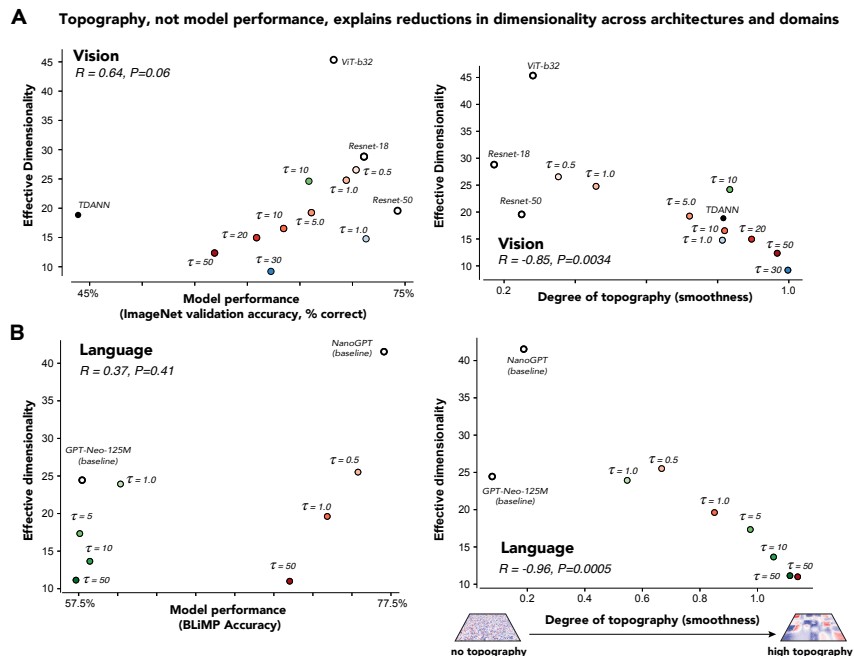

Figure 3: **Topography explains reductions in model dimensionality. A.** (Left) Model performance (Imagenet accuracy, x-axis) versus effective dimensionality for vision ResNets. (Right) Measured topography (smoothness) versus effective dimensionality for vision ResNets. **B.** Same as A, but for language transformers

els (GPT-Neo-125M, NanoGPT). These results are shown in Figure 3A. (Model performance for TDANN is included for comparison, while LLCNN is not reported as it is not yet publicly available.) We found no significant correlation between model performance and dimensionality in either vision or language models (both $P > 0.05$, Figure 3 left column). Notably TopoNets achieved higher model performance despite equivalent levels of model dimensionality as TDANNs (Figure 3A). In contrast, dimensionality was significantly correlated with the measured topography (smoothness) in both domains (each $P < 0.05$). The difference between the linear relationships was statistically significant for both vision and language models ($P < 0.05$, on Fisher's z-transformed correlations).

These results support Hypothesis 2. Spatial topography, rather than model performance, better accounts for reduction in the effective dimensionality of the learned representations. This analysis also shows how TopoNets can be used to evaluate theoretical claims regarding the role of topography in shaping representations across various model architectures and domains.

## 3.3 TOPONETS DELIVER SPARSE, PARAMETER-EFFICIENT REPRESENTATIONS

We next explored a previously unexamined application of TopoNets: model efficiency. Brain-inspired topography encourages compact representations. In biological systems, topographic organization results in localized and redundant information by minimizing "wiring length" (weight sparseness) and enabling more compressible "*metabolically energy-efficient*" representations. Inspired by these biological principles, we asked whether TopoNets, which incorporate similar topographic constraints as brains, might exhibit two forms of efficiency: a) *weight sparseness*, and b) *parameter efficiency*. It is important to clarify that these measures of efficiency are distinct from model dimensionality: effective dimensionality measures the complexity of the feature representation, while weight sparseness and parameter efficiency measure the overall resource use of the model. One concerns the *quality* of the learned features, the other the *quantity* of the resource utilization.

We first assessed weight sparseness in TopoNets by evaluating the effect of pruning small weights using L1 unstructured pruning. Specifically, we set low-magnitude weights to zero and measured

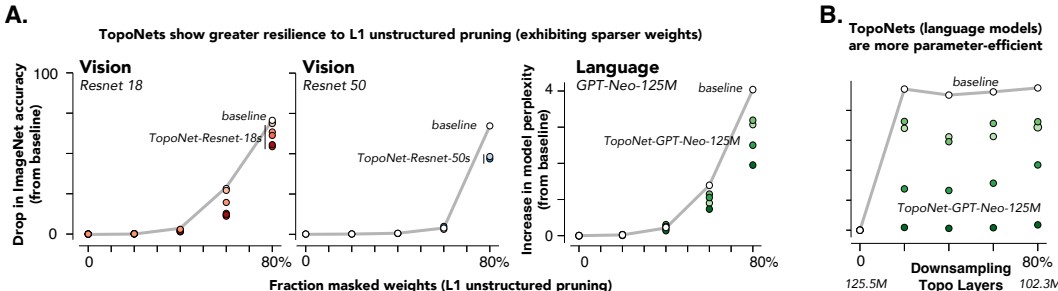

Figure 4: **Measuring the efficiency of TopoNets against baseline models: A.** Fraction of weights masked through L1 unstructured pruning (x-axis) versus the change in model performance (y-axis) for ResNet-18 (left), ResNet-50 (center), and GPT-Neo-125M (right) models. Colored circles represent TopoNets, while hollow black circles represent baseline models. The performance of the baseline models is shown by the gray line. **B.** Percentage of model weights after downsampling (x-axis) versus the drop in model performance (y-axis) for GPT-Neo-125M models.

the impact of this "lesioning" on model performance. We hypothesized that models with inherently sparser weights would be more resilient to L1 pruning. The results, shown in Figure 4A, illustrate the relationship between the fraction of weights lesioned (x-axis) and the corresponding drop in model performance (y-axis). As expected, as the fraction of pruned weights increased, model performance declined. However, across both vision models (ResNet18s and ResNet50s, left and middle subplots) and language models (GPT-Neo-125Ms, right subplot), we found that TopoNets (colored dots) were more resistant to weight pruning than the baseline non-topographic models (black dots). This indicates that TopoNets produce sparser weight distributions and maintain performance more effectively when subjected to L1 unstructured pruning.

However, L1 pruning doesn't directly address the question of parameter efficiency. To test this aspect more directly, we downsampled the weights, thereby directly reducing the model parameter count. Due to architectural limitations, this method works only on transformer models (downsampling convolutional weights results in complete drop in performance to 0). These results are shown in Figure 4B. We found that TopoNets were remarkably resilient to downsampling. For instance, downsampling the weights by 80% lowered the overall parameter count of the model from 125.6M to 102.3M parameters (a 19% overall reduction), while maintaining performance especially at high levels of topography. This shows that TopoNet-GPT models are significantly more parameter-efficient than baseline models. Thus TopoNets offer significant advantages in both weight sparseness and parameter efficiency. The downsampling across both GPT-Neo-125M and NanoGPT (see appendix A.10 and figure 10) results particularly suggest that TopoNets might offer a promising approach to scaling down GPT models without sacrificing task performance. This brain-inspired approach could unlock new methods for compressing large language models, providing a path for more efficient AI systems.

## 3.4 TOPONETS REPRODUCE BRAIN-LIKE TOPOGRAPHIC SIGNATURES

Here we evaluated the "brain-likeness" of TopoNet representations compared to other models. We evaluated vision models on 2 key neural metrics. We first tested unit-to-voxel correlations (as previously reported in Margalit et al. (2024)) from the Natural Scenes Dataset (Allen et al., 2022). Model performance reached the noise ceiling for with comparable prediction accuracies to TDANN models (R = 0.54 for TDANN vs. 0.60 for TopoNet-ResNet-18 and 0.63 for TopoNet-ResNet-50, normalized to the noise ceiling across 8 subjects). Next, we compared TDANNs and TopoNets on neural metrics from BrainScore (Schrimpf et al., 2020; 2018). TopoNets outperformed TDANN at predicting responses across all visual regions (see Table 1 for comparisons between TopoNet and TDANNs, and Appendix A.8 for all TopoNets). Taken together TopoNets predict neural responses on a number of measures better than TDANN. We further replicated key topographic signatures in the visual cortex, such as category selectivity for faces, bodies, and scenes (Kanwisher, 2000; Grill-Spector et al., 2004; Epstein et al., 1999; Downing et al., 2006; 2001), and organizational biases for object size and animacy (Konkle & Oliva, 2011; Konkle & Caramazza, 2013). In Figure 5A, we show these

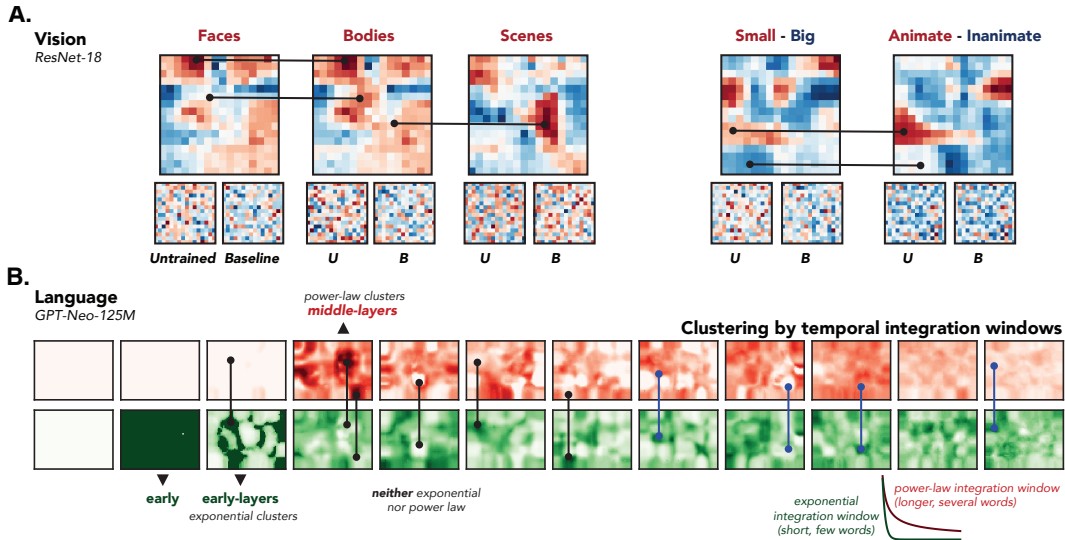

Figure 5: **TopoNets recapitulate topographic signatures observed in the visual and language cortex. A.** Topographic signatures in vision TopoNets (ResNet-18). Colormaps show t-values corresponding to selectivities for faces, bodies, scenes, real-world size, and animacy. Bold connected lines indicate the same regions across different topographic maps. Maps for the same model layer for the untrained (U) and baseline (B) models are shown below for comparison. **B.** Topographic signatures in language TopoNets. Colormaps display the strength of estimated power-law (red), exponential (green), and sentence-yoked coefficients across layers (from left to right). These coefficients indicate fast, slow and sentence-yoked temporal integration windows

patterns for the TopoNet-ResNet-18-$\tau$10 model. We observed that face and body selectivities were yoked together, while scene selectivity was distinct. This pattern mimics the organization observed in the FFA, FBA, and PPA in the ventral visual cortex. We also confirmed this quantitatively. Face and scene selectivity showed a negative correlation (structural similarity = -0.41), whereas face and body selectivity were positively correlated (0.79). Additionally, TopoNets captured similar organizational biases for real-world size and animacy (structural similarity: 0.46), as seen in the brain. Relatively little is known about the spatial organization of the language cortex in the brain, but some

|  | V1 | V2 | V4 | IT |
|---|---|---|---|---|
| TopoNet-ResNet18 | **0.7116** | **0.3038** | **0.2923** | **0.5723** |
| TDANN | 0.6932 | 0.1775 | 0.2792 | 0.4259 |

Table 1: BrainScore values for TopoNet-ResNet-18 and TDANN across visual regions V1, V2, V4, and IT. The scores are averaged over benchmarks (detailed in Appendix A.8)

studies using fMRI (Lerner et al., 2011; Hasson et al., 2008) and invasive recordings (Regev et al., 2024) have provided evidence for distinct temporal receptive fields. Based on these results, we wondered if neurons in TopoNets were clustered by their temporal integration windows. We used a new word-swapping method from a recent study (Skrill & Norman-Haignere, 2024) to investigate this in TopoNets. These temporal integration window results are shown in Figure 5B. We replicated the expected pattern from the previous study: early layers were dominated by exponential integration dynamics, while mid-layers exhibited power-law dynamics. Interestingly, we identified three types of clusters. A) "Exponential" clusters with neurons dominated by short, exponential windows. B) "Power-law" clusters dominated by longer, power-law windows C) An intriguing cluster not explained by either exponential or power-law integration windows. These findings are illustrated in Figure 5B. Topographic maps are presented for all models (including baseline model) in Appendix Figure 8 for comparisons. To our knowledge, this is the first modeling result in topographic language models that recapitulates the experimental findings regarding clusters of temporal receptive field sizes in the language cortex in topographic-LLMs. Further work is needed to establish more

precise correspondences between TopoNets and the human language system, making this an exciting direction for future research.

## 4  DISCUSSION, LIMITATIONS AND CONCLUSION

Here we introduced *TopoLoss*, a novel inductive bias that enables AI models to achieve high task performance while exhibiting topographic organization (like brains). We trained a number of topographic, *TopoNets*, spanning both vision (ResNet-18, ResNet-50, and ViT-b32) and language (GPT-Neo-125M, NanoGPT) domains. TopoNets outperformed prior topographic models on engineering benchmarks while exhibiting comparable topography (Section 3.1), addressed theoretical claims about the importance of topographic principles for low-dimensional feature representations (Section 3.2), delivered parameter-efficient representations (Section 3.3), predicted neural responses better than previous topographic models, and reproduced topographic signatures observed in the brain (Section 3.4).

These results address three central questions about topography and its functional role. *Q1. How* can one incorporate topography in neural networks with minimal drop in task-performance? TopoLoss is a fundamentally novel approach to inducing topography (beyond TDANN and LLCNN) rooted in neuroscientific principles like synaptic pruning. It is an inherently versatile framework that can be applied across model architectures (convolutional networks and transformers) and domains making it a generalizable system for integrating topography into AI systems. *Q2. What* is the representational consequence of brain-like topography? We demonstrate that topography (not task performance) drives representations to be lower dimensional and in turn more brain-like (Section 3.4) representations. This improvement manifests in two ways: (1) improved ability to predict neural data in monkey and human brains (as seen in BrainScore for instance), and (2) in recapitulating key signatures of brain-like processing in the visual and language cortices, such as category-selectivity maps in the visual cortex and temporal integration windows in the language cortex. *Q3. Why* is the brain's design topographic? TopoNets demonstrate parameter efficiency through lesioning (L1 pruning) and downsampling. This offers a novel perspective on the functional significance and evolutionary advantages of topography. This insight underscores the role of topography in optimizing computational systems, providing evidence from a surrogate computational system – artificial neural networks. We also show that TopoLoss integrates seamlessly with fine-tuning methods like LoRA (Appendix A.11). This highlights the complementary nature of TopoLoss and LoRA. TopoLoss imposes topography during training while LoRA enables efficient task-specific fine-tuning. Future work should extend these initial findings into a more comprehensive study of task-specific adaptations, exploring their interaction across a wider range of tasks and model architectures.

**Limitations:** TopoLoss is a versatile framework compatible with foundational ANN components (linear and conv layers). But further work is required to explore the full range of its scalability particularly important in the AI setting. The model backbones in this work (specifically ResNet-18 and GPT-Neo-125M) were chosen to enable comparisons with prior research on topographic models. However, future work will need to scale these models to more complex tasks (beyond ImageNet) and larger architectures (e.g., LLaMA). That said we do not anticipate any challenges in scaling up TopoLoss to more complex architectures. Our model can incorporated in only 2 to 3 lines of additional code (`pip install topoloss`). All preliminary tests show a mere 1-2% performance overhead compared to baseline (non-topographic models). Another limitation is a incomplete understanding of how $\tau$, interacts with model performance and dataset complexity (though see A.4 and Figure 6). A trade-off between topography and model performance is to be expected: if $\tau$ is too high, topography may become overly rigid, limiting the model's ability to learn useful representations. This reflects a well-known principle in computational neuroscience: a critical balance between neural constraints (topographic organization in this case) and task performance. Our framework provides an opportunity to directly test these theoretical ideas in models in future work.

Taken together, TopoNets demonstrate that inducing topographic organization can offer competitive task performance while enhancing the efficiency and interpretability of AI models. This work opens further interdisciplinary work in AI and neuroscience, bringing current AI systems closer to the computational strategies of the brain.

ACKNOWLEDGMENTS

We thank Anna Ivanova and members of the Vision, Cognition, and Computational Lab for feedback on early drafts of the paper. We also appreciate the constructive discussions provided by the four anonymous reviewers. This work was supported by the NIH Pathway to Independence Award (R00EY032603) from the National Eye Institute (NIH) and a startup grant from Georgia Tech (to NARM)

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

## A APPENDIX

### A.1 WHY CHOOSE 2D (AND NOT 3D) CORTICAL SHEETS?

The decision to use 2D cortical sheets to induce topography was rooted in what is known from neuroscience. *First*, the selectivity of neurons in the cortex is known to systematically vary in 2D space, reflecting the organization of feature preferences (e.g., orientation, spatial frequency) across the cortical surface (Kaas, 1997; Katz & Shatz, 1996; Chklovskii et al., 2002; Chklovskii & Koulakov, 2004). Within a cortical column, pyramidal neurons along the depth (third) dimension share relatively similar selectivity, with differences primarily reflecting input/output relationships (Horton & Adams, 2005). As such, topographic organization therefore specifically refers to systematic changes

in selectivity across the 2D sheet. *Second*, our topographic maps are directly based on human fMRI data, which unfolds the brain's 3D folded structure into a 2D cortical sheet. This approach has revealed the organization of the human visual system in great detail (Kanwisher, 2000; 2010; Konkle & Oliva, 2011; Konkle & Caramazza, 2013) and captures the spatial organization of neural activity in a biologically realistic and interpretable manner.

While alternative topographical structures could be of interest for more biological realism, 2D maps allow for a meaningful comparison with prior methods like TDANN (Margalit et al., 2024; Lee et al., 2020), LLCNN (Qian et al., 2024) and provide a neuroscience grounded foundation for studying topography.

## A.2 How to implement the cortical sheet?

The cortical sheet for linear and convolutional layers is implemented by reshaping the weights of the layer into an tensor of shape (`height, width, e`) where `e` refers to the number of input units in linear layers and number of input channels multiplied by kernel size squared in convolutional layers respectively. The components used for implementing the sheet can be found in our source code on github:

- Determining the `height` and `width` of the cortical sheet: link

- Obtaining the cortical sheets for convolutional and linear layers: link

## A.3 Comparing TopoNets, LLCNNs and TDANNs

Each prior approach (LLCNN Qian et al. (2024), TDANN Margalit et al. (2024)) starts from a common base model architecture (ResNet18) and applies an algorithmic modification to induce topography. While this modification could be seen as a change in architecture, the critical question remains: how do these changes impact task performance (e.g., categorization) compared to the unmodified base architecture? Each prior study has highlighted the trade-off between inducing topography and categorization performance, with large reductions in task performance being a recurring concern (despite training for a significantly higher number of epochs).

LLCNN and Toponets are more comparable (trained in a supervised manner on ImageNet). But there are key differences between TDANNs and TopoNets, which make direct comparisons particularly challenging. To ensure transparency for readers, we clearly outline these differences and explain the strategies adopted to facilitate a meaningful comparison between the two models.

*First*, TDANNs were pre-trained on SimCLR and Toponets on Imagenet categorization. The model's training regimen will of course change the model's overall performance. *Second,* TDANN models induce topography on the outputs of the sub-blocks (8 places within the model). In TopoNets we induced topography in every convolutional layer within the ResNet sub-blocks (19 in total). How then do we compare models? We do so in 2 ways.

**Version 1.** We adopt the integrative benchmarking (from BrainScore, Schrimpf et al., 2020) strategy. That is, we compare models on engineering measures (Imagenet) and neural measures irrespective on what they were trained on. Note that even though the comparison between TDANN and TopoNets does not seem fair by this measure (though note that LLCNN and TopoNets can be compared), this is now considered a somewhat accepted measure in vision neuroscience. We present the raw performance of models on Imagenet in Section 3.1 and neural predictivity measures (eg. BrainScore (Schrimpf et al., 2018; 2020)) broken down by visual cortical regions in Section 3.4.

**Version 2.** We report the *difference in model-performance from baseline* (non-topographic versions). For TDANN, the reported difference in categorization performance between the baseline model (trained on SimCLR) and TDANN-SimCLR was 5%. The difference between our baseline model (Imagenet trained) and TopoNet-Imagenet is 7% (for model with all convolutional layers topographic). We trained an additional ResNet-18 model with topography induced on similar layers as TDANN (N=8) for an even more fair comparison. The difference in model performance is 3%. These results are also reported in Section 3.1.

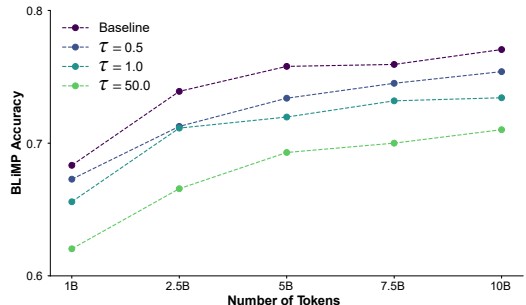

Figure 6: Effect of $\tau$ on on model performance and dataset size. x-axis indicates the number of tokens from FineWeb-Edu used to train a given TopoNet-NanoGPT model and the y-axis indicates the accuracy of the model on BLiMP evaluation. Colors indicate the strength of spatial constraint $\tau$

### A.4 EFFECT OF $\tau$ AND TASK DIFFICULTY ON MODEL PERFORMANCE

$\tau$ controls the degree of topography (spatial constraint) within the model. Theoretically, as topography increases, some drop in model performance is to be expected. Intuitively, a high $\tau$ value (say resulting in a single cluster) would limit the capacity to learn tasks effectively. The brain appears to balance this trade-off by achieving a "sweet spot," optimizing both efficiency and performance. But in order to get further insights we trained several TopoNet-NanoGPT models with increasing numbers of tokens and varying levels of $\tau$ to explore how task performance changes as both data scale and topography constraints vary. These results are shown in Figure 6. In general we observe small performance drops from the baseline model with increasing $\tau$. What is also interesting is that the overall performance drop on the BLiMP dataset does not seem to increase with increasing the number of tokens. Evaluating the effect of topography with increasing model complexity and dataset size is an important area for future research into topographic models. We hope to explore the idea of "Scaling Laws for Topographic Networks" in greater depth in subsequent work.

### A.5 DOWNSAMPLING

To downsample the weights of a linear layer, we first reshape the weight matrix to the cortical sheet (see method for details). The weights are then downsampled along the height and width dimensions by factors $\phi_h$ and $\phi_w$, respectively. The downsampled sheet is then reshaped back to obtain a weight matrix with reduced dimensions: $\left(\frac{n_{\text{output}}}{\phi_h \times \phi_w}, n_{\text{input}}\right)$ where $n_{input}$ and $n_{output}$ are the number of input and output neurons respectively.

Similarly for convolutional layers, we obtain downsampled weights of shape $\left(\frac{o}{\phi_h \times \phi_w}, i, k, k\right)$ where $i$, $o$ and $k$ are the number of input channels, output channels and the kernel size respectively.

The forward pass through this downsampled linear layer proceeds as follows:

- Perform the matrix multiplication between the input tensor (shape: $(\text{batch}, n_{\text{input}})$) and the downsampled weight matrix. The result has shape $(\text{batch}, \frac{n_{\text{output}}}{\phi_h \times \phi_w})$.

- Reshape the result to $(\text{batch}, \frac{c_h}{\phi_h}, \frac{c_w}{\phi_w})$, where $c_h$ and $c_w$ are the height and width of the original cortical sheet, such that $c_h \times c_w = n_{\text{output}}$.

- Upsample the reshaped output by factors $\phi_h$ and $\phi_w$, producing a tensor of shape $(\text{batch}, h, w)$. This upsampled tensor is reshaped back to $(\text{batch}, n_{\text{output}})$.

- Finally, the bias is added to obtain the final output of the downsampled layer.

The forward pass through this downsampled Convolutional layer proceeds as follows:

- Perform the convolution operation between the input tensor of shape: $(\text{batch}, i, \text{height}, \text{width})$ and the downsampled weight matrix. The result has shape $(\text{batch}, \frac{o}{\phi_h \times \phi_w}, \text{height}, \text{width})$

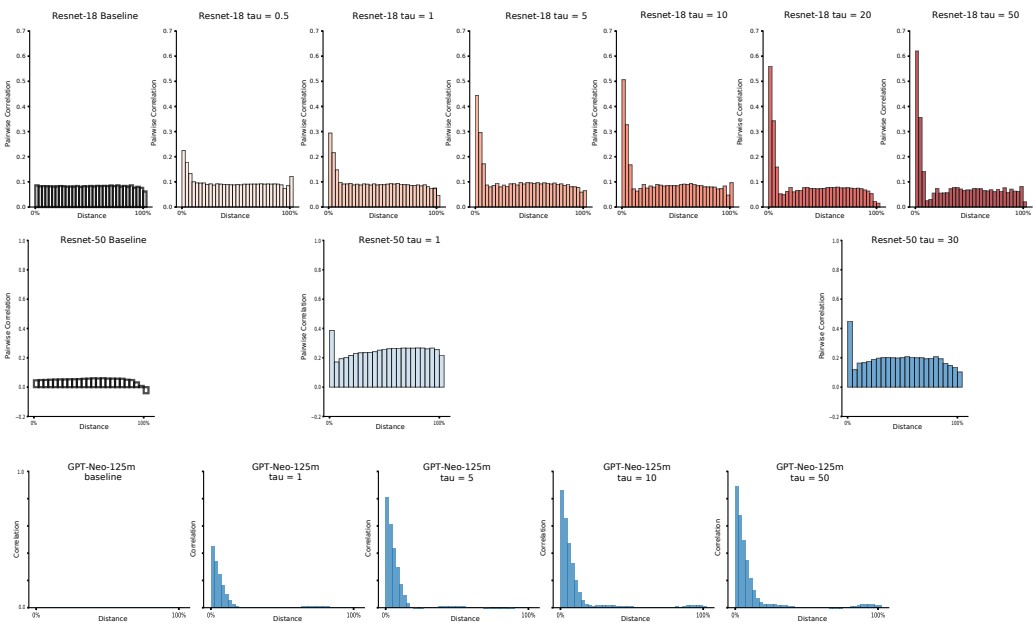

Figure 7: Pairwise Correlation vs. Distance for various models

- Reshape the result to $(\text{batch}, \frac{c_h}{\phi_h}, \frac{c_w}{\phi_w}, \text{height}, \text{width})$, where $c_h$ and $c_w$ are the height and width of the original cortical sheet, such that $c_h \times c_w = o$.

- Upsample the reshaped output by factors $\phi_h$ and $\phi_w$ along the 2nd and 3rd dimensions respectively, producing a tensor of shape $(\text{batch}, c_h, c_w, \text{height}, \text{width})$. The $c_h$ and $c_w$ dimensions are then merged to obtain the output of the convolution operation $o$ output channels.

- Finally, the bias is added to obtain the final output of the downsampled layer.

To reduce the number of parameters in the weights by a factor of $n$, we set $\phi_h = \sqrt{n}$ and $\phi_w = \sqrt{n}$.

## A.6 PAIRWISE DISTANCE VS. CORRELATION

To perform this analysis on the GPT-Neo-125M models, we selected the first 100,000 samples from the `BookCorpus` dataset (Zhu et al., 2015) and extracted the activations from the topographic layers. We then computed the Pearson correlation between the activations of every pair of units within each layer. Finally, for the x-axis, we calculated the Euclidean distance between each pair of units in the cortical sheet space.

For ResNet-18 and ResNet-50, we fed the ImageNet validation set images through the model and collected outputs from the topographic layers. For each of these layers, we computed the Pearson correlation between the outputs of a single channel and all other channels within the same layer. Next, we projected each layer onto the cortical sheet and calculated the Euclidean distance between the corresponding weights of each output channel. Then we plotted the Pearson correlation (y-axis) against the Euclidean distance (x-axis) for the output channels in each layer.

## A.7 TEMPORAL INTEGRATION WINDOWS

We closely followed the source code provided by (Skrill & Norman-Haignere, 2024). The only change that we made was that instead of evaluating the outputs of the `c_proj` layer, we did the same analysis on the topographic layers i.e the `c_fc` layers. Figure 8 visualizes the Power Law and the Exponential parameter estimates for our GPT-Neo-125M models.

A.8   BRAINSCORE BENCHMARKS

We compared TopoNet-ResNet18 models with TDANN in predicting individual neuron responses in the primate visual system using the BrainScore platform. Notably, the $\tau10$ model exhibited smoothness (topography) values most comparable to the original TDANN. Across all regions, TopoNets outperformed TDANN in predicting neural responses.

| Brain Region | V1 | V2 | V4 | IT |
|---|---|---|---|---|
| Baseline | 0.6913 | 0.3038 | 0.2346 | 0.5953 |
| TopoNet-$\tau0.5$ | 0.6906 | 0.1989 | 0.2299 | 0.6334 |
| TopoNet-$\tau1$ | 0.6555 | 0.2664 | 0.2369 | 0.5325 |
| TopoNet-$\tau0.5$ | 0.7262 | 0.1826 | 0.2784 | 0.5722 |
| **TopoNet-$\tau10$** | 0.7116 | 0.3038 | 0.2923 | 0.5723 |
| TopoNet-$\tau20$ | 0.6989 | 0.2614 | 0.3523 | 0.4746 |
| TopoNet-$\tau50$ | 0.6666 | 0.266 | 0.3553 | 0.5369 |
| **TDANN** | 0.6932 | 0.1775 | 0.2792 | 0.4259 |

Table 2: Comparison of BrainScore performance for Baseline, TopoNet (with different $\tau$ values), and TDANN models across brain regions V1, V2, V4, and IT. The scores equate to the mean of multiple benchmarks.

We utilized all BrainScore benchmarks that are publicly accessible through the BrainScore GitHub and AWS. The complete list of benchmarks used is provided below for reference. Note that these include both macaque neural responses and fMRI data.

- **V1:**
    1. Tong.Coggan2024_fMRI.V1-rdm
    2. FreemanZiemba2013public.V1-pls
    3. Marques2020_Cavanaugh2002-grating_summation_field
    4. Marques2020_Cavanaugh2002-surround_diameter
    5. Marques2020_Cavanaugh2002-surround_suppression_index
    6. Marques2020_DeValois1982-pref_or
    7. Marques2020_DeValois1982-peak_sf
    8. Marques2020_Ringach2002-or_bandwidth
    9. Marques2020_Ringach2002-or_selective
    10. Marques2020_Ringach2002-circular_variance
    11. Marques2020_Ringach2002-orth_pref_ratio
    12. Marques2020_Ringach2002-cv_bandwidth_ratio
    13. Marques2020_Ringach2002-opr_cv_diff
    14. Marques2020_Ringach2002-modulation_ratio
    15. Marques2020_Ringach2002-max_dc
    16. Marques2020_Schiller1976-sf_bandwidth
    17. Marques2020_Schiller1976-sf_selective
- **V2:**
    1. Tong.Coggan2024_fMRI.V2-rdm
    2. FreemanZiemba2013public.V2-pls
- **V4:**
    1. MajajHong2015public.V4-pls
    2. Tong.Coggan2024_fMRI.V4-rdm
- **IT:**
    1. MajajHong2015public.IT-pls
    2. Tong.Coggan2024_fMRI.IT-rdm

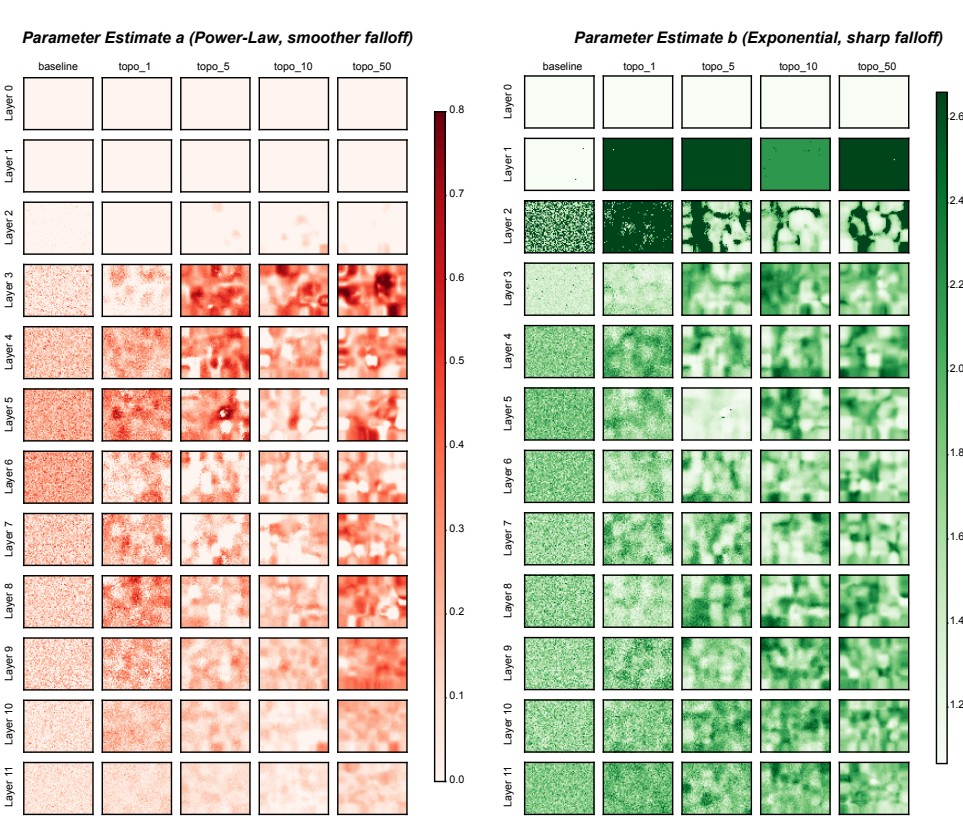

Figure 8: **Temporal integration window estimates for all layers in all of the GPT-Neo-125M models**.

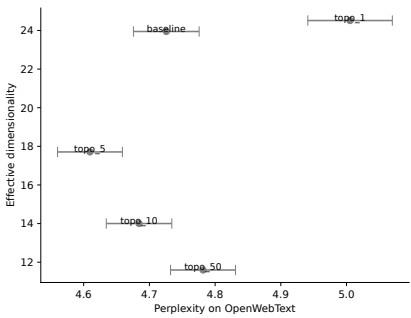

Figure 9: Effective dimensionality of representations v/s model performance for GPT-Neo-125m models trained on Wikipedia.

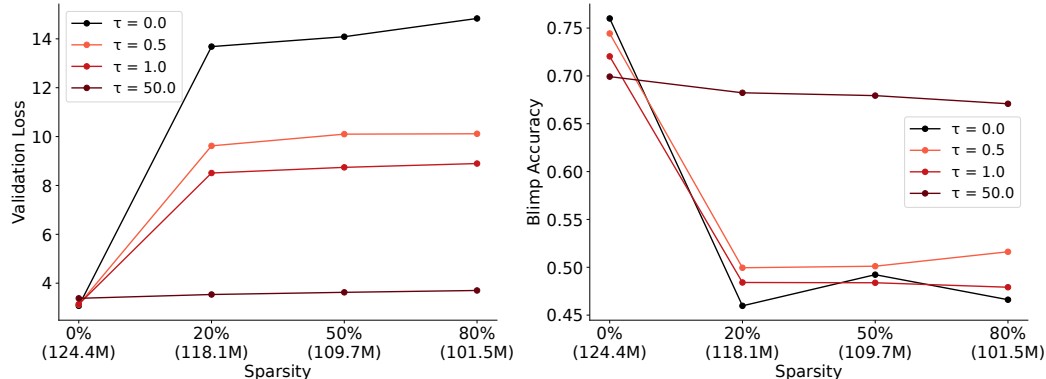

Figure 10: Training NanoGPT with higher $\tau$ significantly improves performance under sparse settings (Downsampling; see Section 3.3) compared to the baseline and models trained with lower $\tau$ values.

## A.9 EFFECTIVE DIMENSIONALITY VS. PERPLEXITY ON OPENWEBTEXT

When we evaluated the GPT-Neo-125M models on the OpenWebText dataset with varying levels of $\tau$, we observed that the effective dimensionality of the layer representations initially increased from the baseline to $\tau = 1$, but then again followed a more predictable pattern from $\tau = 5$ onward (decreasing gradually with increasing perplexity for higher $\tau$ values) - see Figure 9.

## A.10 DOWNSAMPLING TOPOGRAPHIC LAYERS ON NANOGPT

We repeated the Downsampling experiment from Section 3.3 using topographic NanoGPT models, evaluating them on a held-out validation set and the BliMP dataset. The model trained with $\tau = 50$ significantly outperformed (see figure 10) the models trained with smaller $\tau$ values and the baseline model. See Figure 10.

## A.11 THE EFFECT OF TOPOLOSS ON LORA-BASED FINE-TUNING

The purpose of TopoLoss was to shape spatially organized, persistent feature representations within neural networks. In Transformers, this goal was most directly achieved by targeting the mlp.c_fc layer in each transformer block. Prior studies indicate that it is the mlp.c_fc module that encodes "world knowledge" (analogous to what we imagine as the language system in human brains). This makes it the natural and theoretically grounded target for inducing topographic organization. However, this choice raises a potential concern that TopoLoss on the mlp.c_fc module may not align with large-scale transformer models because they employ Low-Rank Adapters (LoRAs, Hu et al. (2021))

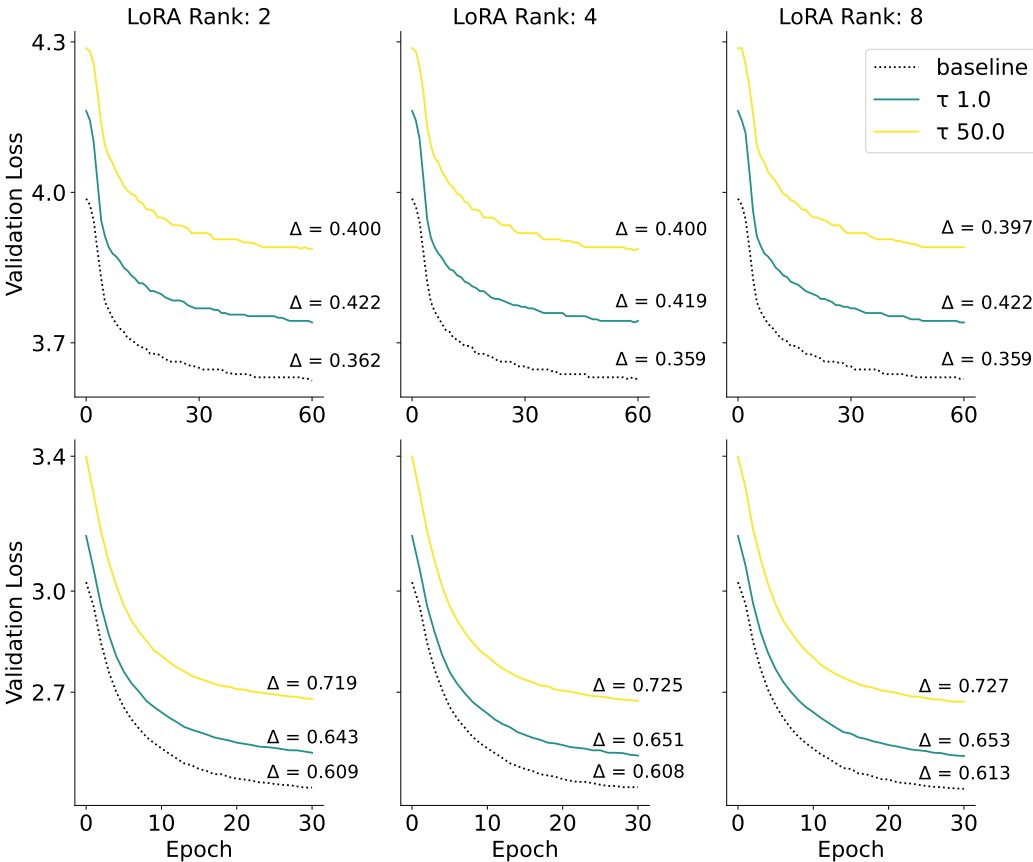

Figure 11: Effect of LoRA fine-tuning on the fully trained NanoGPT checkpoints (both baseline and topographic). The top row corresponds to the Shakespeare dataset and the bottom row corresponds to the Python code dataset.

exclusively on the attention matrix. Here we demonstrate that TopoNets are, in fact, fully aligned and compatible with LoRA fine-tuning.

To validate this compatibility, we fine-tuned our TopoNet-NanoGPT models on two datasets 1. Shakespeare's text (300k tokens) and 2. 1M tokens of Python code (sampled randomly from the tiny-codes dataset Nam Pham (2023)). The experiments were conducted on multiple LoRA ranks (2, 4, and 8) to ensure generalizability. We applied LoRA fine-tuning on our TopoNet-NanoGPT models ($\tau = 1, 50$) and compared the LoRA fine-tuning performance with the baseline NanoGPT model (without topography). These validation loss curves exhibit a characteristic and consistent drop after LoRA-based fine-tuning. Notably, we observe a consistently larger improvement in model performance ($\Delta$) for TopoNet-NanoGPT models compared to the baseline model. These findings, summarized in Figure 12, highlight that TopoLoss integrates seamlessly with LoRA fine-tuning, further enhancing task-specific adaptations while preserving topographic organization.

Together, the results from this additional experiment show that large-scale transformer models trained with TopoLoss can still be fine-tuned via LoRA on the attention matrix. While LoRA is designed to enable efficient task adaptation through fine-tuning, TopoLoss focuses on imposing interpretability and localized representations in the model's persistent representations during pre-training. These objectives are *complementary* rather than conflicting. This combination allows us to leverage the strengths of both methods. LoRA enables efficient and flexible task-specific adaptations, while TopoLoss enhances interpretability, parameter efficiency, and localized representations during pre-training. This division of focus highlights the compatibility and synergy between the two approaches, enabling robust and adaptable models for diverse tasks.

