# OpenReview forum: "TopoNets: High performing vision and language models with brain-like topography"
_ICLR.cc/2025/Conference — ICLR 2025 Spotlight_

### Official Review · Reviewer_FUA2 · 2024-10-26

**Soundness:** 3
**Presentation:** 1
**Contribution:** 3
**Rating:** 8
**Confidence:** 3

**Summary:**

This work explores cortical topography from a representational standpoint, aiming to integrate such in both vision and language model classes to observe a higher performance than previous models that have attempted to do so, while reproducing certain aspects of topographic signatures within the learned feature representations and model parameters. Specifically, TopoNets show category-selectivity for faces, scenes, and bodies. Representations match neural responses in higher visual cortex under one-to-one mapping. Language TopoNets contained clusters of units with distinct temporal integration windows.

**Strengths:**

Modeling cortical topography is an important line of research for both neuroscience and AI, and I appreciate the authors’ work on the matter.
- The authors synthesize the application of a single loss term—namely, TopoLoss—to both residual vision networks and GPT. The motivation behind the loss seems well founded.
- Models trained using TopoLoss incur minimal drop in task performance, which is an important result, since previous models have struggled with this.
- The emergence of specific topographic signatures in sparse networks is again an important finding, given that the brain is optimized to work in an energy-efficient way.

**Weaknesses:**

- Comparisons have been made in the manuscript (such as in result 3.1) where TopoNets are compared to the TDANN. TDANN was trained in a self-supervised way (precisely, using SimCLR), but I did not find (and I apologize if I may have missed it) any mention of the training/finetuning objective (categorization versus self-supervision) that the authors used. If TopoNets were trained through self-supervision, what objective was used? If supervised, is it justified to compare TopoNets with the TDANN? What contribution, if any, do the authors speculate the choice of the training objective plays alongside TopoLoss?
- Figure 6 - I appreciate the analysis around topographic signatures. However, I would prefer seeing the same plots replicated for the baseline vision and language models—(a) without being trained jointly on TopoLoss, (b) unoptimized (control) on both the task and topographic losses. This would really emphasize that what is being shown is not artifactual and attributable to the use of TopoLoss. I hope this is easy enough to check since it does not require any additional training.

**Things (manuscript writing) that can be improved but did not impact my score**:
- In-text citations that are parenthetical and not narrative should be enclosed in parentheses. For example, lines 30-31 should be written as (Barlow, 1986; Rakic, 1988, …). Use the \citep{} LaTeX feature to do so.
- Line 182 - there should be at least a single sentence explanation of what FFCV is.
- “Resnet" -> “ResNet"
- Lines 213+ - the authors explain "L1 unstructured pruning” and “downsampling” as policies but not how they are being used as metrics. This should be made clear at this point in the manuscript.
- The first paragraph of Result 3.1 talks extensively about "model performance” without being explicit about what the task is (I am presuming object categorization) and what dataset it is being evaluated on (i.e., ImageNet). It is only in the figure on the next page that this is made apparent. I would encourage the authors to be more direct in their writing.
- Double quotes, such as on line 398, should be implemented using the ``...’’ LaTeX feature to show both opening and closing quotes.
- There should be a paragraph in the introduction talking about what topographic signatures look like for both vision and language—such as the emergence of pinwheel-like smooth orientation preference maps in V1 (Blasdel and Salama, 1986; Nauhaus et al., 2012), category selective maps in IT/VTC (Desimone et al., 1984), etc. This has been done later in the manuscript, but needs to be brought up earlier.
- Line 450 - "B. opographic" (typo)
- Line 483 - the authors claim that they have created “a broad suite of topographic AI models”. The use of the term “broad suite” is, in my opinion, an overstatement, given that the only model classes that were evaluated were residual networks and GPT. I would rephrase this claim.

**Questions:**

- Line 85: “… show diminished capacity to predict brain data.” (a) "brain data" is a highly vague term - are you implying neural predictivity (and if so, V1? VTC?), image-by-image human behavior prediction, or something else? (b) Margalit et al. (2024) show that TDANNs have a higher NSD voxel correlation to the VTC than purely categorization-driven models under one-to-one matching. Can the authors please clarify?
- Line 161 - “We trained 8 distinct ResNet-18 models …”. The authors mention that one is the baseline, and 6 others are TopoNets. What is the 8th model?

---

> ### Author Response · Authors · 2024-11-20
> **Response 1 to Reviewer FUA2**
>
> > Modeling cortical topography is an important line of research for both neuroscience and AI…
>
> **We thank the reviewer for their constructive feedback and for recognizing the challenges that previous topographic models have faced in maintaining task performance.** Your comments and insights help underscore the importance of addressing this balance, and we hope you will find our results a meaningful progress in this direction. We would like to begin addressing some of the questions raised and share our planned analyses and changes. We will upload an updated text in the next few days and a summary of all the changes. Please let us know if you'd like any other questions/specific changes. Thank you for your engaging with us!
>
> **Q1. Are TDANNs and TopoNets directly comparable?**
>
> **Response:** This is a very important point. We would first like to clarify the key differences between TDANNs and TopoNets which highlights the challenges in making direct comparisons.
>
> 1. TDANNs were trained on SimCLR and Toponets on Imagenet categorization. The model's training regimen will of course change the model's overall performance
> 2. TDANN models induce topography on the outputs of every ResNet sub-block. In TopoNets we induced topography in *every* convolutional layer.
>
> We will update the text to highlight the differences in the model training and layers on which to induce topography.
>
> **Given these differences, how do we compare (benchmark) models?**
>
> **Option 1**: We take a leaf out of **integrative benchmarking** (from BrainScore, Schrimpf et al., 2020). That is, we compare models on engineering measures (Imagenet) and neural measures (V1 and VTC) irrespective on what they were trained on. *We agree* with the reviewer that the comparison between TDANN and TopoNets does not seem fair by this measure (though note that LLCNN and TopoNets can be compared). However, this a somewhat accepted measure in vision neuroscience endorsed even by authors of the TDANN study (senior authors were contributors in the original BrainScore paper, Schrimpf, Kubilius 2018).
>
> **Option 2:** We report the **difference in model-performance** from baseline (non-topographic versions). For TDANN, the reported difference in categorization performance between the  baseline model (trained on SimCLR) and TDANN-SimCLR is 5%. The difference between our baseline model (Imagenet trained) and TopoNet-Imagenet is 7% (for model with all convolutional layers topographic). We also now trained a model with topography on the same layers as Margalit et al (for an even more fair comparison) and the difference in model performance is 3%. We will now report this as well for yet another perspective on task performance.
>
> **We will update the paper to highlight the challenges in comparing models and these new results**
>
> That said, we would very much like the comparison between TDANN and TopoNets on task performance *not* to get in the way of the core contributions. Which is that TopoNets signify **a fundamentally new method for training topographic neural network models that scale well across model architectures and domains (vision and language)** and is strongly supported by all our current (and new results).
>
> **Q2. Figure 6 - I would prefer seeing the same plots replicated for the baseline vision and language models—(a) without being trained jointly on TopoLoss, (b) unoptimized (control) on both the task and topographic losses.**
>
> **Response:** We are happy to provide these. We do not observe clustering of units by category selectivity (vision models) or temporal integration windows (language) in baseline models or unoptimized (control models). We will update the figures to highlight these changes.
>
> **Changes to the paper:** Updated plots with baseline and unoptimized models alongside the results from topographic models.
>
> **Q3. Line 85: “…diminished capacity” (a) are you implying neural predictivity (and if so, V1? VTC?), image-by-image human behavior prediction, or something else?**
>
> **Response:** Sorry that this was vague. We imply neural predictivity in V1 and IT (from BrainScore) as well as neuron-to-voxel match from NSD (see below). We will also present these data in a table to highlight the performance improvements better.
>
> **Q4. (b) Margalit et al. (2024) show that TDANNs have a higher correlations... under one-to-one matching. Can the authors please clarify?**
>
> Response: Yes we also performed the same voxel-to-unit one-to-one on NSD from the VTC for all 8 subjects. We now clarify this better in the text and include a table with these numbers for both TDANN and TopoNets
>
> **Q5. Line 161 - “8 distinct ResNet-18 models …”...What is the 8th model?**
>
> **Response:** Thanks for catching this. This was originally a typo. However, based on the reviewer's feedback we trained an additional ResNet-18 model with topography induced on the same layers as TDANN (see above). So we now do have 8 distinct ResNet18 models.

---

> ### Author Response · Authors · 2024-11-20
> **Response 2 to Reviewer FUA2**
>
> **Thank you for highlighting changes where we could improve the writing and for not letting these things affect your scores**
>
> > In-text citations that are parenthetical and not narrative
>
> We have fixed these now.
>
> > Line 182 - there should be at least a single sentence explanation of what FFCV is.
>
> We expand on FFCV in the text further
>
> > “Resnet" -> “ResNet"
>
> Done
>
> > Lines 213+ - the authors explain "L1 unstructured pruning” and “downsampling” as policies but not how they are being used as metrics. This should be made clear at this point in the manuscript.
>
> We agree. We have changed the text to make this point clearer.
>
> > The first paragraph of Result 3.1 talks extensively about "model performance” without being explicit about what the task is (I am presuming object categorization) and what dataset it is being evaluated on (i.e., ImageNet). It is only in the figure on the next page that this is made apparent. I would encourage the authors to be more direct in their writing.
>
> We agree. We have changed the paper and are now very direct about the task and performance
>
> > Double quotes
>
> Fixed
>
> > There should be a paragraph in the introduction talking about what topographic signatures look like for both vision and language
>
> Excellent point! We introduce topographic signatures right in the introduction now
>
> > Line 450 - "B. Topographic" (typo)
>
> Thank you! Fixed
>
> > Line 483 - the authors claim that they have created “a broad suite of topographic AI models”. The use of the term “broad suite” is, in my opinion, an overstatement, given that the only model classes that were evaluated were residual networks and GPT. I would rephrase this claim.
>
> We feel that "broad suite" is a somewhat fair characterization given the sheer number and variety of TopoNets we introduce in this paper. We have in fact gone significantly over and beyond all previous studies (that typically looked at a single model architecture or domain). But we will now rephrase this to a large number of residual models, vision transformers and GPTs". This adjustment ensure that the description is factually accurate and allows the readers to draw their own conclusions about the breadth of our results and analyses.
>
> **We thank the reviewer again for these suggestions. These changes have made our study more readable and significantly stronger**

---

> > ### Comment · Reviewer_FUA2 · 2024-11-23
> >
> > Thank you for the thorough response.
> >
> > ---
> >
> > > That said, we would very much like the comparison between TDANN and TopoNets on task performance not to get in the way of the core contributions.
> >
> > I find this statement odd, given that one of the *core* contributions, as mentioned repeatedly throughout the abstract and introduction is:
> > - "TopoNets are the highest-performing topographic models to date" (ll. 18-9). Well, this should at least have said, "highest-performing *fully supervised* topographic models".
> > - "We show that TopoNets, our suite of topographic models, outperform all previous topographic models while maintaining similar levels of topography" (ll. 103-4).
> >
> > If the goal of the authors was really to propose a "fundamentally new method for training topographic neural network models", the question becomes why one would prefer using spatial constraints introduced through TopoNets (i.e., "maximizing the cosine similarity between the original cortical sheet and its blurred version") rather than the TDANN (i.e., smoothness of unit responses across the cortical sheet), when both lead to emergent topographic motifs. Maybe there is an explanation here that I am missing.
> >
> > ---
> >
> > > TDANN models induce topography on the outputs of every ResNet sub-block.
> >
> > I do not think that is true. Apart from the very first convolutional layer, units from every layer of the model were embedded in a 2d cortical sheet, and spatial losses were introduced on this retinotopic mapping for each layer. While evaluations were made on specific layers of the model that were most V1-like, V4-like, or VTC-like since comparisons needed to be made to the corresponding visual areas in the ventral cortex, spatial constraints themselves were not limited to these layers (see the Methods section).
> >
> > ---
> >
> > However, having said this, I appreciate the additional experiments the authors are performing, and I would be inclined to raise my score when neural predictivity scores are added to the paper.

---

> > > ### Author Response · Authors · 2024-11-26
> > > **Response 3 to Reviewer FUA2**
> > >
> > > We would really like to thank reviewer FUA2 for engaging with us! We would like address some of the outstanding points raised.
> > >
> > > **I find this statement odd, given that one of the _core_ contributions…**
> > >
> > > We are happy to clarify the intention behind our statement: it was certainly not meant to gloss over specific features or comparisons but rather to highlight other meaningful comparisons (e.g., between TopoNets and LLCNNs) and additional metrics, such as brain predictions. The performance of TopoNets is indeed a core contribution. We wouldn’t be writing this paper and the reviewer not taking our approach seriously if our models did not learn anything meaningful. We have discussed the challenges associated with comparing models trained on different objectives. But performance gains relative to LLCNN are meaningful and undeniable. We do not want to misrepresent the claims though. You will find that we are more careful about saying  _supervised_  models in several places in the updated manuscript.
> > >
> > > **Why one would prefer using spatial constraints introduced through TopoNets?**
> > >
> > > This is an excellent question, and we have both a concise and an extended response.
> > >
> > > **The concise answer (from the BrainScore perspective)**: TopoNets perform better than TDANN on the on neural metrics from BrainScore (updated paper). Prediction accuracies are higher for every brain region we evaluated.
> > >
> > > The more extended answer: TopoNets offer several advantages. (1) Versatility: TopoNets can be applied easily to conv-nets and transformers (2) Ease of implementation: The method is straightforward, requiring only 3 additional lines of code and quick to run (3) Broader applicability: We are making TopoNets available across a number of architectures and domains: ResNet-18s, ResNet-50s, ViT-based in vision, NanoGPTs and GPT-Neo-125M in language.
> > >
> > > **If the goal is to propose a fundamentally new method…**
> > >
> > > We want to emphasize the distinction between the  **observation**—smoothness of unit responses across the cortical sheet—and the  **implementation**  of this property in neural network models. In  **TopoNets**, smoothness emerges via the downsampling mechanism. while in  **TDANNs**, it is achieved via explicit matching with the topographic functional profile and swapping of unit locations. TopoNet is a fundamentally new approach to inducing topography. Both TDANN and TopoNets successfully produce equally smooth unit responses, as demonstrated in our results.
> > >
> > > Indeed there could be infinitely many algorithmic ways of inducing smoothness of nearby model units. Based on the reviewer’s responses, I feel we are in agreement that the litmus test thereafter is with regard to the neural representations learnt. Whether it is on Imagenet (for broader adoption in the AI community), on brain prediction metrics (eg. BrainScore), or how well both models recapitulate the qualitative patterns observed in brains.

---

> > > > ### Author Response · Authors · 2024-11-26
> > > > **Response 4 to Reviewer FUA2**
> > > >
> > > > **I do not think that is true.**
> > > >
> > > > This is easily verifiable information. We have double and triple checked this now. We now provide additional links and details. We want to underscore that it is of utmost importance to us that we do not misrepresent facts.
> > > >
> > > > **Please note that we did not say that TDANN does not incorporate in every layer.**  In the TDANN source code, we can see that there are two objects that are returned from the forward pass: the logits (`flat_outputs`) and the spatial outputs. The spatial outputs is a dictionary containing outputs from the sub-blocks in ResNet18 with names  `layer2.0`  ,  `layer2.1`  etc. This information can be publicly accessed via this link:  [https://github.com/neuroailab/TDANN/blob/80c585df69dcb831d1d6802e0776a8cf25d8cfef/spacetorch/models/trunks/resnet.py#L70](https://github.com/neuroailab/TDANN/blob/80c585df69dcb831d1d6802e0776a8cf25d8cfef/spacetorch/models/trunks/resnet.py#L70)
> > > >
> > > > In the definition of the model training step, we find that that the spatial loss is applied on these spatial outputs i.e in 8 different positions within the model. This information can be publicly accessed via this link:  [https://github.com/neuroailab/TDANN/blob/80c585df69dcb831d1d6802e0776a8cf25d8cfef/spacetorch/train_steps/custom_train_step.py#L75](https://github.com/neuroailab/TDANN/blob/80c585df69dcb831d1d6802e0776a8cf25d8cfef/spacetorch/train_steps/custom_train_step.py#L75)
> > > >
> > > > On the contrary, TopoNet-ResNet18 models have topography induced in  **every conv layer**  (19 in total) in the residual blocks. In order to match the TDANN strategy, we also trained models by applying TopoLoss only to the  `conv2`  module in each sub-block (8 in total). As stated before, we observe a smaller trade-off in performance compared to our original approach (3% drop from baseline, compared to 5% reported for TDANN).
> > > >
> > > > **I appreciate the additional experiments the authors are performing, and I would be inclined to raise my score when neural predictivity scores are added to the paper.**
> > > >
> > > > Thank you for acknowledging our efforts! We have tried to be thorough and careful, supporting each claim with as many lines of evidence as possible. This discussion has been very constructive and reflects a shared commitment to making the paper stronger and better.

---

> ### Author Response · Authors · 2024-11-26
> **Followup 1 to Reviewer FUA2**
>
> We would like to thank reviewer FUA2 for engaging with us! Below we address some of the outstanding points and comments.
>
> **1. Statement about model performance odd, given that one of the *core* contributions…**
>
> **Response:** We are happy to clarify the intention behind our statement: it was certainly not meant to gloss over specific features or comparisons but rather to highlight other meaningful comparisons (e.g., between TopoNets and LLCNNs) and additional metrics. The performance of TopoNets is indeed a core contribution. We wouldn’t be writing this paper and the reviewer not taking our approach seriously if our models did not learn anything meaningful. We have discussed the challenges associated with comparing models trained on different objectives. But performance gains relative to LLCNN are significant and undeniable. We do not want to misrepresent the claims though. You will find that we are more careful about saying supervised models in several places in the updated manuscript.
>
> **2. Why one would prefer using spatial constraints introduced through TopoNets?**
>
> **Response:** This is an excellent question, and we have both a concise and an extended response.
>
> *The concise answer* (from the BrainScore perspective): TopoNets perform better than TDANN on the on neural metrics from BrainScore (updated paper). Prediction accuracies are higher for every brain region we evaluated.
>
> *The more extended answer*: TopoNets offer several advantages. (1) Versatility: TopoNets can be applied easily to conv-nets and transformers (2) Ease of implementation: The method is straightforward, requiring only 3 additional lines of code and quick to run (3) Broader applicability: We are making TopoNets available across a number of architectures and domains: ResNet-18s, ResNet-50s, ViT-based in vision,  NanoGPTs and GPT-Neo-125M in language.
>
> **3. ...a fundamentally new method…**
>
> **Response** We want to emphasize the distinction between the **observation**—smoothness of unit responses across the cortical sheet—and the **implementation** of this property in neural network models. In TopoNets, smoothness emerges via the downsampling mechanism. while in TDANNs, it is achieved via explicit matching with the topographic functional profile and swapping of unit locations. TopoNet is certainly a fundamentally new approach (the implentation) for inducing topography. Both TDANN and TopoNets produce equally smooth unit responses (the observation), as has been quantified and compared in our study.
>
> Now there could be infinitely many algorithmic implementations for inducing smoothness of nearby model units. Based on the reviewer’s responses, I feel we are in agreement that the litmus test for a new topographic model must be with regard to the representations learnt because of this additional spatial constraint. Whether it is on Imagenet (for broader adoption in the AI community), on brain prediction metrics (eg. BrainScore), or how well both models recapitulate the qualitative patterns observed in brains. We tried to cover all of these aspects within our study.

---

> ### Author Response · Authors · 2024-11-26
> **Followup2 to Reviewer FUA2**
>
> **4. I do not think that is true....**
>
> **Response:** We were quite struck by the reviewer's comment. Our response was factually accurate and is easily verifiable.  We took this opportunity to double and triple check this information again in multiple ways. But in order to be sure, we are including additional links and details below. We want to underscore that it is of utmost importance to us that we do not misrepresent anything.
>
> **Please note that we did not say that TDANN does not incorporate spatial loss in every layer.** In the TDANN source code, we can see that there are two objects that are returned from the forward pass: the logits (`flat_outputs`) and the spatial outputs. The spatial outputs is a dictionary containing outputs from the sub-blocks in ResNet18 with names `layer2.0` , `layer2.1`  etc. This information can be publicly accessed via this link: https://github.com/neuroailab/TDANN/blob/80c585df69dcb831d1d6802e0776a8cf25d8cfef/spacetorch/models/trunks/resnet.py#L70
>
> In the definition of the model training step, we find that that the spatial loss is applied on these spatial outputs i.e in 8 different positions within the model. This information can be publicly accessed via this link: https://github.com/neuroailab/TDANN/blob/80c585df69dcb831d1d6802e0776a8cf25d8cfef/spacetorch/train_steps/custom_train_step.py#L75
>
> On the contrary, TopoNet-ResNet18 models have topography induced in *every conv layer* (19 in total) within the residual blocks. In order to match the TDANN strategy, we also trained ResNet18 models by applying TopoLoss only to the `conv2` module in each sub-block (8 in total). As reported before, we observe a smaller trade-off in performance compared to our original approach (3% drop from baseline, compared to 5% reported for TDANN).
>
> **5. I appreciate the additional experiments the authors are performing, and I would be inclined to raise my score when neural predictivity scores are added to the paper.**
>
>
> **Response:**  Thank you for acknowledging our efforts! We strive to be thorough and careful, supporting each claim with as many lines of evidence as possible. We really appreciate the reviewer's time and consideration for increasing the score for our paper.  This discussion has been very constructive and reflects a shared commitment to making the paper stronger and better.

---

> > ### Comment · Reviewer_FUA2 · 2024-11-26
> >
> > Thanks for the thorough response again. That clarifies all of my questions. I have increased my score based on the revisions incorporated in the paper.

---

> > > ### Author Response · Authors · 2024-11-27
> > >
> > > We'd like to thank the reviewer again for updating their assessment of our paper and increasing the score. The paper is significantly stronger because of all the additional feedback.

---

### Official Review · Reviewer_3kin · 2024-10-27

**Soundness:** 3
**Presentation:** 3
**Contribution:** 2
**Rating:** 8
**Confidence:** 3

**Summary:**

The paper addresses the development of a novel loss function designed to create layer-wise 2D topographical organization in NLP and vision models, demonstrating improvements in performance over previous topographic models while maintaining spatial smoothness. This advancement has potential applications in achieving more biologically plausible neural networks with enhanced weight sparsity.

**Strengths:**

The proposed model improves both performance and topographic organization compared to previous models.

The core claim is well-supported, particularly in vision models, with evidence showing better performance and smoothness.

This approach opens up avenues for more biologically grounded AI models and potential advancements in weight sparsity.

**Weaknesses:**

The motivation for choosing a 2D topographic map as the structure is somewhat unclear. Other possible topographical structures might need consideration.

Comparisons lack consistency across architectures, raising questions about fairness (e.g., LLCNN-G has a different architecture).

Claims are primarily substantiated in vision models without equivalent evidence in NLP or other areas. Are there any models to compare in NLP?

In figure 5, the differences in performance between parameter-efficient TopoNets and baseline models only become significant when there are substantial drops in performance.

Specific areas for improvement include:
(1) finding tau values such that TopoNets have an equal smoothness or effective dimensionality to previous models (i.e. showing that with a similar strength of topographic effect, this method works better)
(2) including margin of error in Figure 4

**Questions:**

1.	Are there alternative topographical structures to 2D maps that could be more theoretically justified, and could the cortical sheet be better motivated?
2.	Is there a specific rationale behind comparing this model against LLCNN-G, given the architectural differences?
3.	Why is perplexity lower for tau = 5.0 compared to tau = 1.0, as shown in Figure 4B (Left)? Could adding a margin of error clarify this?
4.	Beyond effective dimensionality and smoothness, what additional metrics could quantify topography?
5.	Would adding a Brain Score comparison with TDANN provide valuable insight into brain-like topographic signatures?

---

> ### Author Response · Authors · 2024-11-19
> **Response1 to reviewer 3kin**
>
> We thank the reviewer for their thoughtful feedback. We are reaching out now to begin addressing some of the questions raised and to share our planned analyses, which we will update in the next couple of days. Thank you for your engagement!
>
> **Q1: Motivation for choosing 2D versus other possible topographic maps.**
>
> **Response:** This is an important point. The choice of a 2D cortical column on which to induce topography was grounded in the understanding of topography from neuroscience. *First*, the selectivity of neurons in sensory cortex is known to change systematically in 2D space. The selectivity reflects the organization of feature preferences (e.g., orientation, spatial frequency) across the cortical surface. The pyramidal neurons along the depth (third) dimension within a cortical column share similar selectivity (but reflect input/output relationships). Therefore topographic organization here specifically refers to systematic changes in selectivity as a function of distance across the 2D cortical sheet.
> *Second*, the topographic maps we describe here are directly based on comparisons to human fMRI maps of the cortical surface. Surface maps unfold the 3D folded brain into a 2D sheet, and capture the spatial organization of neural activity in a biologically realistic manner.
>
> While other topographical structures may hold interest, 2D maps enable us to compare our method with previous approaches (like TDANN and LLCNN) and are the biologically grounded starting point.
>
> **Revision to paper:** We will expand on the motivation for 2D maps and explicitly state the limitations of 2D cortical sheets in the updated manuscript.
>
> **Q2. Is there a specific rationale behind comparing this model against LLCNN-G, given the architectural differences?**
> **Response:** Each prior approach (LLCNN, TDANN) begins with a common base model architecture (ResNet18) and applies an algorithmic modification to induce topography. While this modification could be seen as a change in architecture, the critical question remains: how do these changes impact **task performance** (e.g., categorization) compared to the unmodified base architecture?
>
> LLCNN, especially, was trained in a supervised manner on ImageNet (similar to TopoNet-ResNet18s). Prior studies, including the LLCNN preprint, have highlighted the trade-off between inducing topography and categorization performance, with large reductions in task performance being a recurring concern (despite training of significantly higher number of epochs).
>
> It is within this context, our results are meaningful and significant. **TopoNets** represent an algorithmic approach to inducing topography that achieves substantially higher task performance while maintaining equivalent (or higher) levels of topography. This increase performance is critical because it helps a theoretical debate regarding the relationships between dimensionality, topography, and task performance (Section 3.2 and Figure 4).
>
> **Proposed Change:**
> 1. We will clearly highlight the challenges in comparing TopoNets with LLCNN and TDANN, particularly emphasizing the architectural changes and task performance differences, to provide greater transparency in the comparisons
> 2. We will explicitly report the **change in model performance** (after inducing topography) relative to the base model, emphasizing how TopoNets outperform both the unmodified architecture and previous topographic models, making the improvements undeniable.
> 3. We will emphasize how TopoNets’ performance dismantles lingering doubts about the trade-off between topography, dimensionality, and task performance.
>
> **Q3: Claims are primarily substantiated in vision models without equivalent evidence in NLP. Are there any models to compare in NLP?**
>
> **Response:** The reviewer is correct that some of the claims are substantiated in vision models. This is largely because topography in language models is in its infance. At the time of submission, there were no alternative approaches for introducing topography into language models during model training. Note that we did note performance of Topoformers (Figure 3). This makes our work foundational in this area. That said, we would like to highlight some of the core claims of our study that do generalize across vision and language.
>
> **Claim 1:** That TopoLoss can be integrated into vision and language models (this is the first paper that applies and compares a topographic loss across conv-nets and transformers in a single study)
>
> **Claim 2:** That TopoNets are high-performing across vision and language models (Figure 3). We are now training NanoGPT models to further strengthen the claim in the language domain.
>
> **Claim 3:** That topography **not** task performance explains reductions in effective dimensionality of the learned representations in TopoNet models (Figure 4).
>
> **Claim 4:** That TopoNets recapitulate brain-like signatures (category-selectivity maps in vision and temporal integration windows in language)

---

> > ### Author Response · Authors · 2024-11-19
> > **Response2 to reviewer 3kin**
> >
> > **Q4: In figure 5, the differences in performance between parameter-efficient TopoNets and baseline models only become significant when there are substantial drops in performance.**
> >
> > **Response:** This observation is correct and aligns with what we report.  This result should be interpreted as compressibility—how well a model can maintain task performance when its parameter budget is reduced. TopoNets demonstrate a unique ability to retain relatively high performance even under highly constrained conditions with limited parameters, where baseline models experience sharp declines. This suggests that TopoNets may be effectively finding the “lottery tickets” in a more structured and efficient way. We will expand on this concept in the revised manuscript to highlight its significance and broader implications for efficient representation and robust performance in neural networks. Thank you for raising this point!
> >
> > **Q5: Specific areas for improvement include: (1) finding tau values such that TopoNets have an equal smoothness or effective dimensionality to previous models (i.e. showing that with a similar strength of topographic effect, this method works better) (2) including margin of error in Figure 4**
> >
> > **Response to (1) - Matching Smoothness or Effective Dimensionality:** This is precisely what we aimed to address with \tau = 10. For example, when \tau = 10, the smoothness for TDANN is 0.816, and for TopoNet-RN18, it is 0.82, allowing for a direct comparison under similar levels of topographic effect. Unfortunately, for LLCNN, this is more challenging because the model is not yet publicly available, and we had to rely on data extracted from the paper which is also in the same ballpark (0.79).
> >
> > **Response to (2) - Including margin of error:** We share the reviewers’ intuition that the difference in perplexity scores is most likely due to random fluctuations and that including margins of error would improve clarity and interpretability. We will update Figure 4 in the revised manuscript to include error bars, providing a more robust presentation of the results.
> >
> > **Q6. Beyond effective dimensionality and smoothness, what additional metrics could quantify topography**
> >
> > **Response:** Beyond dimensionality and smothness, one could come up with many other metrics to quantify topography. For example, sparseness:
> >
> > $S = \frac{\left( \frac{1}{n} \sum_{i=1}^n r_i \right)^2}{\frac{1}{n} \sum_{i=1}^n r_i^2}$
> >
> > Sparseness captures how sharply the selectivity correlation (r) drops as function of distance.
> > We carefully decided to rely on previously used measures like smoothness rather than introducing new metrics, as inventing a novel measure could be perceived as arbitrary and unvalidated. This makes our reported findings interpretable, robust, and comparable to prior studies. This choice also ensures that the focus remains on the insights provided by the model, rather than on the validity of a completely new metric.
> >
> > **Q7: Would adding a Brain Score comparison with TDANN provide valuable insight into brain-like topographic signatures?**
> >
> > **Response:** This is great suggestion. Unfortunately TDANN is not currently available on BrainScore. We are now evaluating TDANN and TopoNets on BrainScore and will update the manuscript with these results shortly.
> >
> > **We thank Reviewer 3kin again for their feedback. We will get back once all the changes have been incorporated. In the meantime, please feel free to let us know if there are any additional suggestions or points you’d like us to address. Thank you!**

---

> > > ### Comment · Reviewer_3kin · 2024-11-21
> > > **Response to authors**
> > >
> > > I thank the authors for their responses. All of my questions are answered. I will update the rating once the changes are incorporated.

---

> > > > ### Comment · Reviewer_3kin · 2024-11-26
> > > >
> > > > I thank the authors for including the proposed revisions. I have raised my rating.

---

> > > > > ### Author Response · Authors · 2024-11-26
> > > > > **Thank you!**
> > > > >
> > > > > We would like to thank the reviewer for engaging with us, for acknowledging our efforts, and for updating their assessment of our paper and increasing the score. The paper is undoubtedly much stronger because of all the additional feedback.

---

### Official Review · Reviewer_gnkR · 2024-11-03

**Soundness:** 3
**Presentation:** 3
**Contribution:** 3
**Rating:** 6
**Confidence:** 4

**Summary:**

The paper introduces a novel loss function called TopoLoss, which promotes spatially organized topographic representations in AI models without compromising task performance. The authors present TopoNets, a suite of models that incorporate this loss function into existing architectures, such as ResNet-18, ResNet-50 for vision tasks, and GPT-Neo-125M for language tasks. The key contributions include demonstrating that TopoNets outperform previous topographic models while maintaining high performance and replicating the topographic signatures observed in the human brain's visual and language cortices, thus bridging biological and artificial systems.

**Strengths:**

The strengths of the paper lie in its innovative approach to integrating topographic organization into AI models through the introduction of TopoLoss. This loss function effectively promotes spatially organized representations, which are crucial for achieving localized feature processing and lower dimensionality, like biological neural networks. The experimental validation across diverse architectures, including ResNet-18, ResNet-50, and GPT-Neo-125M, showcases the adaptability of TopoLoss and its ability to enhance model performance without sacrificing accuracy.

**Weaknesses:**

The weaknesses include a lack of extensive benchmarking against state-of-the-art models beyond those already tested, which may limit the generalizability of the findings. While TopoLoss is shown to improve performance, the paper does not sufficiently address how it scales with larger models or more complex tasks, raising questions about its applicability in high-dimensional settings. Additionally, the discussion surrounding the choice of scaling factor τ could be more detailed, particularly regarding its impact on various architectures and datasets. In addition, the inspiration from the brain's vision and language processing is superficial. The authors could strengthen their claims by providing additional insights into potential trade-offs between topographic organization and task-specific performance metrics!

**Questions:**

How does the choice of scaling factor τ affect performance across varied datasets beyond ImageNet?

Can you elaborate on the potential limitations of TopoLoss when applied to larger or more complex architectures?

What specific challenges do you encounter while integrating TopoLoss into existing architectures, particularly transformers?

---

> ### Author Response · Authors · 2024-11-21
> **Response 1 to Reviewer gnkR**
>
> We thank the reviewer for their thoughtful feedback. We are reaching out now to begin addressing some of the questions raised and to share our planned analyses, which we will update in the next couple of days.
>
> **Q1. The weaknesses include a lack of extensive benchmarking against state-of-the-art models beyond those already tested, which may limit the generalizability of the findings.**
>
> The model backbones used in this study—ResNet-18 for vision and GPT-Neo-125M for language—were selected to enable direct comparisons with previous studies on topographic models in human neuroscience. Please note we did try to move beyond architectures by enabling topographic versions of Resnet50 and GPT-Neo-125M which are more expressive model architectures. In this sense, TopoLoss is a novel algorithm specifically designed to introduce brain-like topography. The key comparison in this paper is between previous methods for inducing topography and our proposed TopoLoss, not SOTA models. We kindly request the reviewer to evaluate the paper within this intended context, particularly its application to address theoretical questions and its relevance to neuroscience. TopoLoss signifies an advance in brain-inspired topographic modeling across architectures and domains.
>
> **Q2. While TopoLoss is shown to improve performance, the paper does not sufficiently address how it scales with larger models or more complex tasks, raising questions about its applicability in high-dimensional settings.**
>
> **Response:**  Scalability is in fact a key strength of TopoLoss which in retrospect, we should have emphasized more and presented further evidence for. As we noted above, we needed to provide evidence that our algorithm works by comparing it with prior methods that exclusively used ResNet-18 architectures.
>
> We thank the reviewer for encouraging us to showcase this aspect further. To address scalability explicitly, we are now training:
>
> 1.  Vision Transformers (`vit_b_32`) on ImageNet
> 2.  NanoGPTs
>
> We will share the updated results in the manuscript and hope these additional tests to more complex architectures (Vision transformers) and tasks (NanoGPTs on 10B tokens) will allay the reviewer’s concern regarding the scalability of TopoNets in high dimensional domains. That said, we kindly request the reviewer to also reconsider the neuroscience context of our work. TopoNets are designed not only for scalability but to provide insights into brain-inspired principles and their computational consequences, bridging gaps between neuroscience and AI. We hope these updates address the reviewer’s concerns fully. Thank you!
>
> **Changes to the paper.**  We will update the paper with more scaled up models (as above).
>
> **Q3. Inspiration from the brain's vision and language processing**
>
> **Response:**  We are tremendously inspired by the human brain. This study is squarely in the space of developing novel algorithms that endow current models with more brain-like principles to understand (1)  _how_  to incorporate brain-like mechanisms into AI systems, (2)  _what_  the representational consequences of brain-like constraints are, and (3)  _why_  the brain is designed the way it is. Our study directly addresses these foundational questions in the following way.
>
> 1.  _How_  to incorporate topography in neural networks? TopoLoss is a fundamentally novel approach to inducing topography rooted in neuroscientific principles (and Turing patterns, for afficionados). Beyond resulting in high-performing TopoNets, our algorithm is an inherently versatile framework that can be applied across model architectures (conv-nets and transformers) and domains.
> 2.  _What_  is the representational consequence of brain-like topography? Topography drives representations to be lower dimensional (Section 3.2) compared to baseline models. Consequently results in more brain-like (Section 3.4) representations. This manifests in two ways: (1) by improving the ability to predict neural data in monkey and human brains , and (2) in recapitulating signatures of brain-like processing in the visual and language cortices (like category-selectivity maps and temporal integration windows).
> 3.  _Why_  is the brain’s design topographic: TopoNets demonstrates parameter efficiency through lesioning (L1 pruning) and downsampling. This provides a new perspective on the role of topography in a surrogate computational system (ANNs), offering insights into its functional significance and evolutionary advantages.
>
> **Changes to the paper.**  We will now refactor the Discussion to directly articulate the neuroscience inspiration and the relevance of our work with these points.

---

> > ### Author Response · Authors · 2024-11-21
> > **Response 2 to Reviewer gnkR**
> >
> > **Q4. How does the choice of scaling factor τ affect performance across varied datasets beyond ImageNet?**
> >
> > **Response:**  This is an excellent and important question.  **τ**  controls the degree of topography (spatial constraint) within the model. Theoretically, as topography increases, some drop in model performance is expected. Intuitively, a high τ value (e.g., resulting in a single cluster) limits capacity to learn tasks effectively. The brain appears to balance this trade-off by achieving a “sweet spot,” optimizing both efficiency and performance.
> >
> > But we want to directly address the authors’ question. We submit that model training on larger vision datasets beyond ImageNet is unfortunately not feasible given the time limitations over the rebuttal period and the compute limitations in academic settings. However, we will be addressing this question in the context of language models. We are currently training several  **nanoGPT models**  with increasing numbers of tokens and varying levels of  **τ**  to explore how task performance changes as both data scale and topography constraints vary. These experiments will provide an initial picture of the relationship between τ and performance at scale. We acknowledge this as an important area for future research and hope to explore the idea of “Scaling Laws for Topographic Networks” in greater depth in subsequent work. For now, we hope the reviewer considers our study within its intended neuroscientific context. We appreciate the reviewer’s understanding and will share the results of our ongoing experiments in the revised manuscript. Thank you for raising this thought-provoking point!
> >
> > **Q5. Can you elaborate on the potential limitations of TopoLoss when applied to larger or more complex architectures?**
> >
> > TopoLoss is designed to work on the foundational components of ANNs i.e linear and Conv layers. This makes it easily compatible with larger and more complex architectures developed on the similar canonical components. We do not foresee any technical issues in applying it to larger or more complex architectures. Based on all current tests, we also anticipate that models trained with TopoLoss will outperform prior strategies for incorporating topography in terms of task performance and predicting brain responses in visual and language cortex
> >
> > As for a prescriptive guidance, we note that some performance trade-off with topography is natural and to be expected (see discussion above). For example, if the scaling factor \tau is set to be too high, the spatial constraints could become overly rigid, preventing the model from learning sufficiently diverse or useful representations. If the scaling factor \tau is set to be too high, the spatial constraints could become overly rigid, preventing the model from learning sufficiently diverse or useful representations. This trade-off reflects a well-known principle in computational neuroscience: balancing topographic organization with task performance. Our framework provides an exciting opportunity to test these theoretical ideas directly in models, and we will expand on this point in the revised manuscript.
> >
> > **Changes to the paper**: We will include a section on model performance versus training data size for varying levels of $tau$ (see above) and on the limitations of topography.
> >
> > **Q6 What specific challenges do you encounter while integrating TopoLoss into existing architectures, particularly transformers?**
> >
> > So far, we have not encountered any significant challenges in integrating this loss into transformers (both ViT and GPT). With our source code (see Appendix) integrating topoloss into any existing training loop requires only 2 extra lines of code and a very small performance overhead (1-2% slower training in our settings and no extra GPU memory usage). We will of course open-source all our code and make it available as a pip installable library. In our experience, it is easier to incorporate TopoLoss into transformers than convolutional networks.
> >
> > So far we have not encountered any inherent limitations with the integration process itself. As we outline above, we have started training scaled up models (as requested by the reviewer). Our training loss and topoloss seem to be proceeding as expected. We will update the reviewer with more results soon.
> >
> > **Changes to the paper**: We will show scaled up models on more complex architectures (see above). Will also include a section on the limitations of Topoloss clearly articulating the challenges.
> >
> > We thank Reviewer gnkR again for their thoughtful feedback. We will revert with updates once we have incorporated our planned changes. In the meantime, please don’t hesitate to share any additional feedback, request clarifications, or raise points you’d like us to address. These points have made our paper stronger. We value the opportunity for further interaction and look forward to continuing the discussion. Thank you!

---

> > > ### Author Response · Authors · 2024-11-27
> > > **Updates to paper based on suggestions from reviewer gnkR**
> > >
> > > Dear Reviewer gnkR,
> > >
> > > As we haven’t yet received additional feedback during this discussion period, we’re kindly reaching out to request your review of the updated version of the paper to ensure that all the concerns you raised have been addressed.
> > >
> > > You will find that we have made substantial updates in response to your thoughtful feedback, including the following specific changes:
> > >
> > > 1. **Scaled up models:** We now include results for Vision Transformers (ViTs) and NanoGPTs to demonstrate the scalability of TopoLoss to more complex architectures and tasks. (Response to Q1 and Q2)
> > > 2. **Brain inspiration and contributions:** We have expanded the discussion on our inspiration from the brain and explicitly detailed the core contributions of this work to our understanding of neural systems. (Response to Q3)
> > > 3. **Additional experiments on $\tau$:** We have conducted new experiments to explore how $\tau$ affects task performance across varied datasets. These results are included in the updated manuscript. (Response to Q4)
> > > 4. We have added an explicit section discussing the limitations of TopoLoss, as requested. (Response to Q5)
> > > 5. We outline the practical challenges encountered while incorporating TopoLoss into existing model training pipelines, providing clarity for future work. (Response to Q6)
> > >
> > > Additionally, we have carefully refined the manuscript, providing more precise claims, updated figures, and deeper explanations to address all points raised in the review.
> > >
> > > We hope these revisions comprehensively address your concerns and provide clarity on the contributions of our work. If you find the updates satisfactory, we kindly request you to consider updating your scores to reflect the improvements made. Please let us know if there are any remaining issues or additional suggestions. Thanks for your time and for considering our paper!

---

> > > > ### Comment · Reviewer_gnkR · 2024-11-28
> > > > **response to rebuttal**
> > > >
> > > > I thank the authors for providing satisfactory responses to my concerns. I have raised my rating.

---

> > > > > ### Author Response · Authors · 2024-11-28
> > > > >
> > > > > We’d like to thank the reviewer once again for updating their assessment of our paper and increasing the score. We greatly value your feedback. We'd be happy to address any additional comments or concerns you might have.

---

### Official Review · Reviewer_bq7J · 2024-11-04

**Soundness:** 3
**Presentation:** 3
**Contribution:** 3
**Rating:** 8
**Confidence:** 3

**Summary:**

This paper introduces a novel topoloss aimed at aligning AI models with the structure of brain neurons to create brain-like topography within these models. The topoloss defines a cortical sheet within AI models by reshaping the weight matrix and maximizes the cosine similarity between this cortical sheet and its blurred version, simulating synaptic pruning in the brain. The authors apply the topoloss to both CNN for vision and transformer for language models. Evaluation results indicate that the resulting toponets achieve a balance between maintaining topographical structure and overall model performance.

**Strengths:**

The idea of mapping model weights to a topographical cortical sheet is interesting, with a simple and elegant approach. The reshaping technique appears versatile and could potentially be applied to the weights of various models. The discussion on topographic signatures within toponets highlights promising avenues for brain-inspired AI research.

The paper is well-written, and the idea is easy to follow. The authors have provided sufficient detail and references to support reproducibility.

**Weaknesses:**

The selected backbone is somewhat outdated and lacks current relevance, reducing the overall significance of the work. For vision and language models, more meaningful choices would include ViT and LLMs like Llama.

No performance improvement compared to the original no-topo model, nor is there any discussion on time and data efficiency or interpretability. This makes the topography alignment appear theoretical, aligning with brain structures only on a formulaic level without contributing to human-level intelligence.

**Questions:**

Consider incorporating ViT and LLaMA as backbones for the framework to enhance relevance and performance.

Compare the application of topological loss on the transformer's attention matrix instead of the fc matrix.

---

> ### Author Response · Authors · 2024-11-20
> **Response 1 to bq7j**
>
> **Q1:  The selected backbone is outdated, and more meaningful choices like ViT or LLaMA should be considered.**
>
> **Response:**  The model backbones used in this study—ResNet-18 for vision and GPT-Neo-125M for language—were carefully selected to enable direct comparisons with previous studies on topographic models in human neuroscience. Specifically, ResNet-18 has been used in prior work as a standard architecture for inducing topography in vision models (LLCNN and TDANN). Similarly, GPT-Neo-125M has demonstrated strong alignment with human brain data in language processing tasks, as evidenced by results on benchmarks like BrainScore-Language.
>
> We interpret the reviewer’s question as probing whether TopoNets can scale to larger, more complex architectures.  **They absolutely can!**
>
> To demonstrate the scalability of TopoNets, we have begun training on larger and more contemporary architectures, including ViT (as suggested by the reviewer) and nanoGPT (on 10B tokens from fineweb-edu). Early results already confirm that TopoNets scale effectively, maintaining their performance and topographic properties.
>
> **Planned Experiments:**  We are training Topographic ViTs and nanoGPTs to further demonstrate the scalability, generalizability, and relevance of TopoNets for modern, larger-scale architectures. Results from these experiments will be included in the revised manuscript.
>
> **Q2:  The method does not demonstrate performance improvement or discuss efficiency or interpretability.**
>
> **Response:**  The notion of performance improvement in our work is within the context of incorporating  _brain-like principles_  into standard neural networks. In this sense, TopoLoss is a novel and generalizable algorithm specifically designed to introduce brain-like topography. The key comparison in this paper is between previous methods for inducing topography and our proposed TopoLoss, not SOTA models. We kindly request the reviewer to evaluate the paper within this intended context, particularly its application to address theoretical questions and its relevance to neuroscience. TopoLoss signifies an advance in brain-inspired topographic modeling across architectures and domains.
>
> As we demonstrate, TopoNets outperform previous topographic models on engineering performance metrics (esp LLCNNs where the comparisons are more directly meaningful, see Reviewer 4) and even at predicting responses in the brain (compared to TDANNs).
>
> Regarding performance improvements with baseline models on neuroscience inspired benchmarks
>
> 1.  We show that TopoNets are lower dimensional (like brains) compared to baseline (non-topographic) models despite their high task performance (Section 3.2)
> 2.  TopoNets are more efficient in terms of parameter usage and weight sparseness compared to baseline models, demonstrating representational economy. (Section 3.3)
> 3.  Toponet feature maps are more directly interpretable and brain-like compared to baseline models (Section 3.4 and updated Figure 6 with category-selective and temporal integration window maps for baseline and untrained models)
>
> These findings underscore the dual achievement of maintaining task performance while incorporating biologically inspired principles into neural nets. Thank you for the opportunity to clarify this!
>
> **Q3.  Consider incorporating ViT and LLaMA as backbones for the framework to enhance relevance and performance.**
>
> **Response:**  We have started training two models with TopoLoss:
>
> 1.  ViTs on Imagenet
> 2.  NanoGPTs on the FineWeb-Edu dataset
>
> We will update the paper with these results soon.

---

> ### Author Response · Authors · 2024-11-20
> **Response 2 to bq7j**
>
> **Q4.  Compare the application of topological loss on the transformer's attention matrix instead of the fc matrix.**
>
> **Response:**  We appreciate the reviewer’s suggestion and would like to respectfully clarify why applying topographic loss to the transformer’s attention matrix may not align with the intended goals of our approach. The purpose of topographic loss is to shape spatially organized, persistent feature representations, and in transformers, this is most directly achieved by targeting the  `mlp.c_fc`  layer in each transformer block. Prior studies (Geva 2020, Geva 2022) indicate that it is the `mlp.c_fc`module that encodes “world knowledge” (analogous to what we imagine as the language system in human brains). This makes it the natural and theoretically grounded target for inducing topographic organization. The attention layers, by contrast, maps the interactions along the sequence dimension (i.e between tokens). It does not store persistent feature representations, which limits its suitability for the kind of spatial organization we aim to achieve.
>
> That said, we greatly value the reviewer’s insights and wonder if the suggestion may be intended to explore a different or complementary aspect. If so, we would be more than happy to address this point further with additional context or clarification. Thank you for raising this thought-provoking question!
>
> **Changes to the paper:**  We will better motivate the reason for inducing topography in  `mlp.c_fc`  module as opposed to the attention matrix.

---

> > ### Comment · Reviewer_bq7J · 2024-11-25
> >
> > Thanks for the additional experiments and explanations. I’m happy to raise my score after the revision.

---

> > ### Comment · Reviewer_bq7J · 2024-11-26
> > **6: marginally above the acceptance threshold**
> >
> > Thank you for the revision. I have raised my rating to 6.
> >
> > My only remaining concern is that the proposed topo loss on the mlp.c_fc module of the transformer may not align well with large-scale transformer models, as these models typically employ LoRA fine-tuning exclusively on the attention matrix.

---

> ### Author Response · Authors · 2024-11-27
> **Response to remaining concern raised by Reviewer bq7J**
>
> We really appreciate the reviewer's time and consideration for increasing the score for our paper!  **We do wish to address the reviewer's remaining concern.**
>
> **Alignment of TopoLoss with LoRA in large-scale models**
>
> We greatly appreciate the reviewer’s thoughtful feedback and the additional context regarding the interaction of large-scale models and LoRA fine-tuning. This is a deeply insightful point and highlights an important aspect of TopoLoss that we would like to clarify.
>
> *LoRA* is designed to enable efficient task adaptation through fine-tuning, while *TopoLoss* is focused on imposing interpretability and localized representations in the model during pre-training. These objectives are *complementary* rather than *conflicting*. Investigating the intersection of LoRA’s low-rank adaptations with TopoLoss would require extending the study to task-specific adaptations. I hope the reviewer concurs that doing so is beyond the scope of the current paper.
>
> **But we want to provide evidence that TopoLoss causes no issues with LoRA fine-tuning.** Large-scale transformer models trained with TopoLoss can still be fine-tuned via LoRA on the attention matrix. This division of focus allows us to leverage the strengths of both methods:
>
> **LoRA:** For efficient and flexible task-specific adaptations.
>
> **TopoLoss:** For interpretability, parameter efficiency, and localized representations during pre-training.
>
> To further support this claim, we are now running an additional experiment demonstrating that TopoLoss and LoRA can be used together. Specifically, TopoLoss can be imposed on the mlp.c_fc module during pre-training, while LoRA can be applied to the attention matrix during fine-tuning. We will include the results in the revised manuscript to further clarify this point.
>
> Thank you for raising this important concern, and we look forward to presenting these findings asap.

---

> ### Author Response · Authors · 2024-11-28
> **Compatibility of TopoLoss with LoRA fine-tuning**
>
> We have now updated the paper once more to specifically address Reviewer bq7J’s remaining concern.
>
> **Q1. TopoLoss on the mlp.c_fc module of the transformer may not align well with large-scale transformer models, as these models typically employ LoRA fine-tuning exclusively on the attention matrix.**
>
> **Response:** We have now updated the paper with the results from the additional experiments (we proposed yesterday) demonstrating that **models trained with TopoLoss align seamlessly with LoRA fine-tuning** on the attention matrix. Specifically, we evaluated our TopoNet-NanoGPT models, fine-tuned using LoRA on two datasets and across three LoRA ranks (to ensure robustness and generalization of our findings). The results from this experiment are now included in **Appendix A.11** and in an **additional paragraph in the Discussion**.
>
> We find that the validation loss for fine-tuning curves show a  characteristic and consistent drop following LoRA-based fine-tuning (as expected). Notably, TopoNet-NanoGPTs consistently achieve a larger improvement in model performance compared to the baseline model (likely because of ceiling effects).  These findings, summarized in Figure 12, provide important empirical support of our previously theoretical claim: that **TopoLoss integrates seamlessly with LoRA fine-tuning.**
>
> We hope these additional experiments that we’ve now conducted will address reviewer’s remaining concern. We kindly request a renewed consideration of our study in light of these additional experiments and updated results. Thank you again for raising this question and for the opportunity to address the remaining concerns.

---

### Author Response · Authors · 2024-11-27
**Global Response 1**

Dear Reviewers,

We have now uploaded an updated version of the paper. These changes reflect a considerable additional effort to address core conceptual choices, clarify issues with model comparisons, and include a significant number of additional TopoNet models to demonstrate the generalizability of TopoLoss to more complex datasets and model architectures.

Key changes include:

1. **New models to demonstrate generalizability:**  We have now included a) TopoNet-ViTs trained in a supervised manner, and b) TopoNet-nanoGPTs trained on 10B tokens from FineWeb-Edu (6 additional models: 4 TopoNets, 2 baselines). All the figures have been updated to reflect these changes. The results demonstrate that TopoNets can in fact be scaled easily. Importantly, all the core contributions remain as-is (if at all are significantly strengthened with additional evidence)
2. **BrainScore:** The updated paper includes benchmarking of TopoNets and TDANNs on the neural metrics from BrainScore. These comparisons are included both in Table1 and the Appendix. TopoNets outperform TDANN on neural metrics for every brain region available on the BrainScore platform.
3. We have included the rationale for choosing **2D (versus 3D) topographic maps**. These changes are presented briefly in the Methods and an extended version is included in the Appendix.
4. **Control and Baseline Maps:** We have now included topographic maps for the baseline (non-topographic model) and random model to directly facilitate comparisons between models.
5. **Stated claims more carefully.** We have been more precise in our stated claims about model  performance.
6. **Effect of task complexity wrt Tau on model performance.** We have included additional experiments to investigate the effect of task complexity and spatial constraints on model performance. This is presented in the Appendix.
7. **Efficiency plots for NanoGPTs** are now added in the Appendix. As before, models trained with a high $\tau$ show significantly smaller degradation in performance under sparse settings (when compared to baseline).
8. We have now included an explicit **section on the limitations of TopoLoss** and discussed the challenges in incorporating TopoLoss to existing architectures.
9. We have improved the **explanation for explicit inspiration from the brain’s vision and language processing**. This now includes a paragraph in the discussion on the broad questions addressed by our study.

Thank you again for your consideration. These changes have undoubtedly made the paper significantly stronger. We are now performing some more additional analyses regarding the interaction of LORAs and TopoLoss. We will update with further changes as soon as possible.

---

### Meta-Review · Area_Chair_yLzY · 2024-12-21

**Metareview:**

The work proposes TopoNets, a framework for integrating brain-inspired topography into AI models using the novel TopoLoss, demonstrating spatially organized representations without sacrificing performance. Strengths include its innovative approach, comprehensive validation across vision and language models, and successful generalization to diverse architectures like ResNet, ViTs, and GPT-Neo. Weaknesses raised included limited benchmarking against state-of-the-art models and insufficient analyses of topography's impacts, but the authors addressed these with additional experiments, improved explanations, and explicit scalability tests. The strong empirical results, robust theoretical grounding, and interdisciplinary relevance to both AI and neuroscience suggest for an acceptance.

**Additional Comments On Reviewer Discussion:**

During the discussion, the reviewers highlighted concerns about scalability, benchmarking against state-of-the-art models, and the clarity of certain claims regarding the benefits of TopoLoss. The authors addressed these by conducting additional experiments with Vision Transformers (ViTs) and NanoGPTs to demonstrate scalability, providing direct comparisons to baselines, and improving the manuscript with clearer explanations of trade-offs and topographic impacts. They also added analyses of neural predictivity and incorporated suggestions for clarity in figures and results. The authors’ detailed and timely responses, along with robust experimental evidence, addressed most concerns, raising the reviewer scores.

---

### Decision · Program_Chairs · 2025-01-22

Accept (Spotlight)